# Rethinking Loss Reweighting for Imbalance Learning as an Inverse Problem: A Neural Collapse Point of View

**Jinping Wang** [* 1 2] **Zixin Tong** [* 2] **Zhiwu Xie** [2] **Zhiqiang Gao** [† 1 2]

## Abstract

Loss reweighting is a widely used strategy for long-tailed classification, but existing reweighting strategies often rely on heuristics and rarely define a well-specified target. Inspired by Neural Collapse (NC), the ideal simplex Equiangular Tight Frame (ETF) terminal geometry suggests equal per-class average loss as a reasonable target for reweighting. Based on the ideal equal loss objective, we consider loss reweighting as an inverse problem and propose an inverse-view reweighting strategy that infers class weights dynamically to match this ideal objective. Empirically, NC metrics suggest our method can effectively reduce the loss imbalance coefficient and achieve closer alignment with NC geometry while consistently outperforming strong long-tailed baselines on different datasets. Our code is publicly available at: https://github.com/tongzixin716716/Inverse-Loss-Reweighting.

## 1. Introduction

Deep Neural Networks have achieved huge success on various visual recognition tasks (Krizhevsky et al., 2012; He et al., 2016). These achievements are largely supported by large-scale datasets such as ImageNet (Deng et al., 2009), where each class has an equal and balanced number of training samples. However, in real-world scenarios, datasets are rarely perfectly balanced but often follow a long-tailed distribution. In a long-tailed distribution, head classes have a large number of samples and dominate the sample space, while the tail classes only have a few samples, causing the model to over-bias towards the head classes.

*Equal contribution , †Corresponding author. [1] International Frontier Interdisciplinary Research Institute, Wenzhou-Kean University [2] CSMT, Wenzhou-Kean University. Correspondence to: Zhiqiang Gao <zgao@wku.edu.cn>, Jinping Wang <1306325@wku.edu.cn>, Zixin Tong <1338028@wku.edu.cn>, Zhiwu Xie <zxie@wku.edu.cn>.

*Proceedings of the 43rd International Conference on Machine Learning*, Seoul, South Korea. PMLR 306, 2026. Copyright 2026 by the author(s).

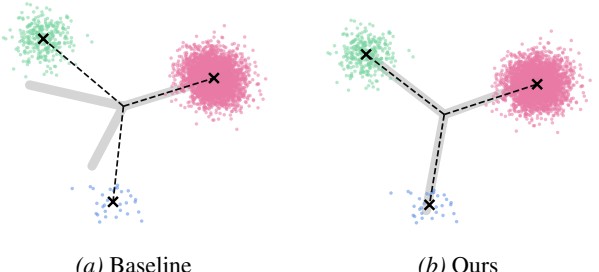

*(a)* Baseline      *(b)* Ours

*Figure 1.* Toy examples with 2-dimensional features and 3 classes to illustrate the feature learning of classification. Black crosses are the class means, and gray lines are the classifier weights.

Aiming to improve the generalization ability and learn better representations under such imbalanced scenarios, diverse long-tailed methods have emerged (Cui et al., 2019a; Menon et al., 2020; Kang et al., 2019). For example, data resampling strategies (Han et al., 2005) mitigate class-imbalance bias through reshaping the sample classes dominating the decision boundary by learning the separation representation from classifier optimization. Contrastive learning strategies have also been applied to long-tailed scenarios (Kang et al., 2020), by pulling positives together and pushing apart negatives, models are able to learn more discriminative and uniformly structured embeddings.

Among all the imbalance learning strategies, loss reweighting is one of the most widely used paradigms (Cui et al., 2019a; Lin et al., 2017; Park et al., 2021). By simply adjusting the class-wise weights, loss reweighting strategies enable more balanced training, leading to better generalization performance under imbalance training scenarios. However, current reweighting methods are often target-agnostic: Instead of setting an ideal class-wise loss distribution, reweighting is often designed as a heuristic. Thus, we take a step back and rethink: what ideal objective should reweighting aim for?

To make the above question concrete, we need an explicit description of the classifier at the end of training. The phenomenon of Neural Collapse (NC) provides such a description: At the terminal phase of training, within-class features collapse to class means, the class means form an

ideal Equiangular Tight Frame (Simplex ETF) while the classifier weights align accordingly, yielding a highly symmetric last-layer geometry. Under NC geometry, such an ideal symmetric configuration will force all classes to have an equal average loss. This points to a direct design principle for reweighting strategy: reweighting should explicitly drive class-wise loss equally at the end of training.

Based on this principle, we propose to view long-tailed reweighting as an inverse problem. With class-wise equal loss as the ideal objective, we formulate a Tikhonov-regularized inverse problem, obtain a closed-form per-class solution of the optimal loss weight for each class, and implement it in a batch-wise plug-and-play manner. Extensive experiments verify the effectiveness and the validity of our reweighting strategy. Specifically, the observation of the NC metric demonstrates our method can better recover the ideal ETF geometry under a long-tailed scenario while consistently outperforming other reweighting strategies under different imbalance ratios. Meanwhile, when working with other long-tailed approaches, our method can consistently outperform the original methods and achieve the state-of-the-art (SOTA) performance. Our main contribution can be summarized as follows:

- We identify the ideal target missing in loss reweighting for long-tailed training. Current reweighting strategies rarely define an ideal loss distribution they aim to reach, making the design heuristic.

- We connect NC with long-tailed reweighting and set class-wise equal loss as an ideal end-state objective. We show that loss imbalance obstructs convergence to an NC-consistent solution, motivating class-wise equal loss as an ideal objective.

- We cast reweighting for long-tailed learning as an inverse problem and derive a plug-and-play algorithm. With the loss-balancing target, we formalize an inverse objective with a per-class closed-form solution.

## 2. Preliminaries

### 2.1. Notation

The given training set consists of $C$ classes is imbalanced, and our goal is to train a model that can generalize well on balanced sets. Each class contains $n_c$ samples and each sample can be denoted as $\{(x_{i,c}, y_{i,c})\}$ where $x_{i,c} \in \mathbb{R}^d$ denotes the $i_{th}$ input sample of class $c$ and $y_{i,c} = c$ denotes its real label. We consider the layers before the classifier as a feature extractor that the mapping $h: \mathbb{R}^d \to \mathbb{R}^p$ outputs a $p$-dimensional feature vector $h(x)$. A linear classifier with weight matrix $\mathbf{W} \in \mathbb{R}^{C \times p}$ and biases $b \in \mathbb{R}^C$ takes the last-layer features as inputs and then outputs the class label. Specifically, through classification scores via

$f(x) = \mathbf{W}h(x) + b$, the predicted label is then given by $\arg\max_{c'}\langle w_{c'}, h \rangle + b_{c'}$, where $w_{c'}$ denotes the classifier weight for a specific class. Furthermore, we denote the class mean $\mu_c = \frac{1}{n_c}\sum_{i=1}^{n_c} h(x_{i,c})$, the global mean $\mu_G = \frac{1}{C}\sum_{c=1}^{C}\mu_c$ and the centered class mean $\hat{\mu}_c = \mu_c - \mu_G$.

For a sample $(x_{i,c}, y_{i,c})$ the logits and the softmax probabilities can be defined as:

$$z(x_{i,c}) := Wh(x_{i,c}) \in \mathbb{R}^C,$$
$$p_k(x_{i,c}; W) := \frac{\exp(z_k(x_{i,c}))}{\sum_{j=1}^{C}\exp(z_j(x_{i,c}))}. \qquad (1)$$

Let $\ell$ be any label-symmetric per-sample loss:

$$\ell(W, h(x_{i,c}), c) := \psi(z(x_{i,c}), c),$$
$$where \ \ z(x_{i,c}) := Wh(x_{i,c}). \qquad (2)$$

The class-wise average loss of class $c$ is:

$$L_c(W) := \frac{1}{n_c}\sum_{i=1}^{n_c}\ell(W, h(x_{i,c}), c). \qquad (3)$$

### 2.2. Simplex ETF

A Simplex Equiangular Tight Frame (Simplex ETF) in $\mathbb{R}^p$ is defined as a set of $C$ vectors obtained from the columns of a matrix $\mathbf{M} \in \mathbb{R}^{p \times C}$. A general representation of this matrix is

$$\mathbf{M} = \sqrt{\frac{C}{C-1}}\,\mathbf{R}\left(\mathbf{I} - \frac{1}{C-1}\mathbf{1}_C\mathbf{1}_C^{\top}\right), \qquad (4)$$

where $\mathbf{I} \in \mathbb{R}^{C \times C}$ is the identity matrix and $\mathbf{1}_C \in \mathbb{R}^{C \times 1}$ is the vector of ones. The matrix $\mathbf{R} \in \mathbb{R}^{p \times C}$ is an orthogonal rotation satisfying $\mathbf{R}^{\top}\mathbf{R} = \mathbf{I}$. Thus, the resulting matrix $\mathbf{M} = [\,m_1, m_2, \dots, m_C\,]$ gives $C$ class-specific vectors in $\mathbb{R}^p$, with each column $m_c$ representing the weight associated with class $c$.

### 2.3. Neural Collapse

As Papyan (Papyan et al., 2020) demonstrated, at the terminal phase of the training for balanced datasets, the last-layer feature will converge to the class means, aligning with the classifier weights and converging to a symmetric Simplex ETF structure. This phenomenon is called Neural Collapse (NC) and can be formally described in the following four phases:

**(NC1) Within-Class Variability Collapse:** As training continues, the intra-class variation of activations diminishes, causing the activations of samples from the same class to

collapse toward their class means. Specifically, it means the within-class covariance matrix $\Sigma_W$ approaches zero:

$$\Sigma_W = \frac{1}{Cn} \sum_{c=1}^{C} \sum_{i=1}^{n} \left(h(x_{i,c}) - \mu_c\right) \left(h(x_{i,c}) - \mu_c\right)^\top,$$
$$\Sigma_W \to 0, \tag{5}$$

**(NC2) Convergence to a Simplex ETF:** The mean vectors of each class will converge to a simplex ETF. Let $\dot{\mu}_c = (\mu_c - \mu_G)/\|\mu_c - \mu_G\|_2$ denotes the re-normalized class means, the NC2 process can be described as:

$$\|\mu_c - \mu_G\|_2 - \|\mu_{c'} - \mu_G\|_2 \to 0, \quad \forall c, c',$$
$$\langle \dot{\mu}_c, \dot{\mu}_{c'} \rangle \to \frac{C}{C-1}\delta_{c,c'} - \frac{1}{C-1}, \quad \forall c, c', \tag{6}$$

where $\delta_{c,c'}$ denotes the Kronecker delta symbol.

**(NC3) Convergence to Self-Duality:** The classifier weight $w_c$ will gradually aligned with the corresponding centered class mean $\dot{\mu}_c$. The NC3 process can be described as:

$$\left\| \frac{\mathbf{W}^\top}{\|\mathbf{W}\|_F} - \frac{\dot{\mathbf{M}}^\top}{\|\dot{\mathbf{M}}\|_F} \right\|_F \to 0, \tag{7}$$

where $\dot{M} = [\dot{\mu}_c, \ c = 1, ..., C] \in \mathbb{R}^{p \times C}$.

**(NC4) Simplification to Nearest Center** Given a feature, the neural network classifier converges to the nearest class mean. The NC4 process can be described as:

$$\arg\max_{c'}\langle w_{c'}, h \rangle + b_{c'} \ \to \ \arg\min_{c'} \|h - \mu_{c'}\|_2. \tag{8}$$

# 3. Ideal Objective of Reweighting Strategies For Imbalance Learning

## 3.1. Missing Objective For Current Reweighting Strategies

Under a long-tailed distribution, the standard cross-entropy loss is unable to generalize well since the model tends to over-fit the head classes while neglecting the tail classes. To alleviate this, a variety of reweighting strategies are proposed, including static or adaptive sample frequency-based reweighting strategies (Cui et al., 2019a; Lin et al., 2017) and some advanced dynamic strategies based on the feature space (Zhang et al., 2017) or decision boundaries (Park et al., 2021) and meta-learning approaches (Jamal et al., 2020). More details of the mentioned reweighting approaches are shown in the Appendix A.

However, existing reweighting approaches are target-agnostic, and they do not specify what the class-wise loss

distribution should look like, resulting in heuristic ways to design different reweighting strategies. To address this pitfall, we introduce the phenomenon of Neural Collapse (NC) which describes the ideal geometric behavior of a well-trained Neural Network at the terminal phase of training. Based on the insights given by NC, we derive and analyze an ideal loss configuration in the following section.

## 3.2. Neural Collapse Inspired Ideal Objective

Under balanced training, at the terminal phase of training, the Neural Network will exhibits NC phenomenon where the classifier weights will gradually align with the class means and converge to an ideal geometry of Simplex ETF. Thus, many imbalanced learning approaches are trying to recover such a phenomenon with different strategies. Under NC, all classes are geometrically symmetric in the last-layer representation at the end of training. With such an ideal property, the following theorem shows that this symmetry forces their average losses to be identical.

**Theorem 3.1.** *Assume NC1, NC2, and NC3 when the centered class means are aligned with the classifier weights and converged to a Simplex ETF. Then every class has the same class-wise average loss:*

$$L_1(W) = L_2(W) = \cdots = L_C(W). \tag{9}$$

*Proof.* Shown in Appendix B.1 ☐

Theorem 3.1 characterizes what an ideal class-wise average loss distribution looks like under Neural Collapse at the terminal phase of training: once the features and the classifier weights have aligned and converged to a simplex ETF, all classes necessarily share the same average loss. However, for long-tailed training scenarios, the class-wise losses are typically imbalanced. This raises a natural question: *Can the network still converge to an ideal simplex ETF if such a loss imbalance persists?*

Recall the per-class average losses $L_c(W)$, we have the global mean $\bar{L}(W) = \frac{1}{C}\sum_{c=1}^{C} L_c(W)$. We define the *class-wise loss imbalance coefficient* by

$$\rho(W) := \frac{\sqrt{\frac{1}{C}\sum_{c=1}^{C}\left(L_c(W) - \bar{L}(W)\right)^2}}{\bar{L}(W)},$$
$$\text{whenever } \bar{L}(W) > 0. \tag{10}$$

By definition $\rho(W) \geq 0$, and $\rho(W) = 0$ if and only if $L_1(W) = \cdots = L_C(W)$, where all classes have exactly the same average loss. The next theorem precisely discusses that as long as $\rho(W)$ is not equal to zero, the model will not converge to an ideal ETF structure.

**Theorem 3.2** (Loss imbalance precludes convergence to ETF). *Let $\{W_t\}, t \geq 0$ denote the sequence of classifier*

*parameters produced by training at time step t. For each t, let $L_c(W_t)$ be the class-wise average losses and let $\rho(W_t)$ be the loss imbalance coefficient. Assume that:*

1. *There exist constants $\varepsilon > 0$ and $T \in \mathbb{N}$ such that :*

$$\rho(W_t) \geq \varepsilon, \qquad \forall\, t \geq T.$$

    *Where the loss for each class remains imbalanced.*

2. *The sequence $\{W_t\}$ admits at least one limit point in parameter space.*

*Then any limit point $W_\star$ of $\{W_t\}$ cannot be an ETF solution satisfying NC1-NC3. In particular, if $\{W_t\}$ converges to some $W_\dagger$, then $W_\dagger$ is not an ETF solution.*

*Proof.* Shown in Appendix B.2 □

Theorem 3.2 illustrates that loss imbalance is the fundamental obstruction that prevents the neural network from reaching the ideal NC geometry under a data imbalance setting. This observation naturally leads to the central objective of our reweighting method: **to enforce class-wise average loss balance**. The next section introduces our approach that explicitly targets this goal by treating reweighting as an inverse problem.

# 4. Proposed Method

## 4.1. Reweighting as an Inverse Problem

As we demonstrated in the previous analysis, to recover the ideal ETF structure of the Neural Collapse process, at the terminal phase of training, each class should have balanced loss where the imbalance factor $\rho(w)$ should converge to 0. Conversely, if $\rho(w)$ remains bounded away from zero, the centered class mean and the classifier weights will fail to converge. This gives a very concrete target to design the reweighting loss strategy:

**Target:** To move long-tailed training toward an NC-consistent regime, a reweighting mechanism should dynamically adjust the effective contribution of each class so that their average losses become as balanced as possible, actively driving the imbalance factor $\rho(w) \to 0$ at the end stage of training.

Most reweighting methods for long-tailed learning adopt a forward view: they directly prescribe a class weight $\{w_c\}_{c=1}^C$. However, we argue that, with the ideal reweighting target mentioned above, **we can take an inverse view instead**. Specifically, we treat reweighting as an inverse problem: with the ideal balanced class-wise losses objective, we then infer the underlying class weights that produce this behavior. Formally, we can define the following Tikhonov-regularized inverse problem:

**Problem 1.** *Given the current network parameter W, consider the class-wise losses $L_c(W)_{c=1}^C$ and their mean $\bar{L}(w)$. The ETF condition suggests that, ideally, each effective class loss should have an equal loss $\bar{L}(W)$. Given $\alpha$ as a scalar, we therefore propose to obtain the class weights $\{w_c\}$ by solving the following inverse problem*

$$\arg\min_{\{w_c\}} \sum_{c=1}^C \left(w_c L_c(W) - \bar{L}(W)\right)^2 + \alpha \left(w_c - w_c^{(0)}\right)^2 \tag{11}$$

In Eq. 11, the first term encourages loss equalization which encourages $w_c L_c(W)$ towards the ideal target $\bar{L}(W)$. The second term is an optional Tikhonov regularizer that keeps $w_c$ close to a prior weight $w_c^{(0)}$ from the original method (if available, otherwise $w_c^{(0)} = 1$), inferred weights remain compatible with the original method.

Notably, the objective in Eq. 11 is separable over classes. Thus, we can define the following class-wise inverse problem:

**Problem 2.** *For each class c, we can obtain the class weights $w_c$ by solving the following optimization problem:*

$$\arg\min_{w_c} \left(w_c L_c(W) - \bar{L}(W)\right)^2 + \alpha \left(w_c - w_c^{(0)}\right)^2. \tag{12}$$

The separable quadratic admits a closed-form solution. Thus, we have the following theorem for per-class optimal weights.

**Theorem 4.1** (Closed-form solution for per-class optimal weights)**.** *Then the objective in Eq. (12) is strictly convex and admits a unique minimizer*

$$\boxed{w_c^\star(W) = \frac{\bar{L}(W)\, L_c(W) + \alpha\, w_c^{(0)}}{L_c(W)^2 + \alpha}.} \tag{13}$$

*Proof.* Shown in Appendix B.3 □

## 4.2. Batch-Wise Inverse Reweighting

Based on the inverse formulation in the previous section, we now turn it into a practical training algorithm. In practice, our method performs a batch-wise inverse reweighting strategy: for each training batch, we look at the classes that actually appear in this batch and solve the inverse problem shown in Problem 2 with the closed form in Eq. 13 to obtain the batch-wise weight $w_c$ for each class.

However, due to the class-frequency skew, different classes appear in very different numbers of mini-batches throughout training, where head classes appear in almost every batch while the tail classes are observed sporadically. As a result,

even though the tail classes are properly reweighted whenever they appear, their cumulative optimization strength over the entire training process remains significantly weaker compared to the head classes. To alleviate this issue, we further introduce a macro-level reweighting strategy in the next section.

### 4.3. Macro-Level Reweighting

Since the effective contribution of a class depends not only on its loss weight within a batch, but also on how frequently it appears in optimization across batches, the optimization strength for head classes is significantly higher than that of the tail classes. To address this, we introduce a macro-level, batch-frequency-aware reweighting mechanism, which complements our batch-wise inverse formulation.

Let $B_c$ denote the number of mini-batches that class $c$ appears during training (we track this quantity online that increments $B_c$ by one whenever class $c$ appears in a batch). We then define the macro reweighting factor for each class as follows:

$$\beta_c \propto (B_c)^{-\gamma}, \tag{14}$$

where $\gamma \geq 0$ controls the strength of the macro compensation. To avoid the global loss scale, the macro reweighting weights $\beta_c$ are normalized to have a unit mean. Thus, we can have the final effective class weight $\hat{w}_c$:

$$\hat{w}_c = \beta_c \cdot w_c^\star \quad , \tag{15}$$

where $w_c^\star$ is the batch-wise class weight obtained from the closed form in Theorem 4.1. The full algorithm of our method is shown in the Algorithm 1.

## 5. Experiments

In this section, we evaluate our proposed method on long-tailed benchmarks. All experiments are conducted on NVIDIA RTX 4090 GPUs, and the results are reported as the mean over three random seeds. **Experiment details can be found in Appendix F**.

### 5.1. Performance Comparison With Other Reweighting Methods

To evaluate the effectiveness of our reweighting strategy, we compare our algorithm with other long-tailed reweighting methods. We train the model for 200 epochs using ResNet-32 (He et al., 2016) as backbone on CIFAR-100-LT (Cui et al., 2019b) and compare the following eight different reweighting loss strategies: (i) standard cross-entropy (CE), (ii) inverse-frequency weighting (Inv-Freq), (iii) inverse-square-root weighting (Inv-Sqrt), (iv) class-balanced reweighting (CB (Cui et al., 2019a)), (v) range reweighting (Range (Zhang et al., 2017)), (vi) two-component reweighting (TCR (Jamal et al., 2020)), (vii)

focal reweighting (Focal (Lin et al., 2017)), and (viii) influenced-balanced reweighting (IB (Park et al., 2021)). All hyper-parameters were either tuned to their optimal values or set according to the recommendations in the original paper. Further details of the above eight methods and experiments can be found in Appendix F.

We evaluate our proposed reweighting method with eight other reweighting methods, and we show the performance in Table 1. We conduct this experiment on the CIFAR-100-LT dataset under three imbalance factors (IF=50, 100, 200). In the table, we can see that our proposed reweighting method achieves the best performance under all three imbalance ratios and improves 7.08%, 6.26%, and 6.36% points from the cross-entropy baseline, respectively.

### 5.2. The Evolution of NC Metrics Across Different Reweighting Strategies

In this subsection, we analyze how the NC metrics evolve during training under different reweighting strategies. Following prior work (Zhou et al., 2022), we compute the three neural collapse metrics NC1-NC3 on the last-layer features and the classifier to quantify the first three NC properties. The experimental settings in this section are consistent with the previous section 5.1.

By using the notation we defined before, we introduce the first three NC metrics as follows:

**NC1:** The within-class and between-class covariance matrices are

$$\Sigma_W := \frac{1}{n_c C} \sum_{c=1}^{C} \sum_{i=1}^{n_c} \big(h(x_{i,c}) - \mu_c\big)\big(h(x_{i,c}) - \mu_c\big)^\top \in \mathbb{R}^{p \times p},$$

and

$$\Sigma_B := \frac{1}{C} \sum_{c=1}^{C} \big(\mu_c - \mu_G\big)\big(\mu_c - \mu_G\big)^\top = \frac{1}{C} \sum_{c=1}^{C} \hat{\mu}_c \hat{\mu}_c^\top \in \mathbb{R}^{p \times p}.$$

With $\Sigma_B^\dagger$ the pseudo inverse of $\Sigma_B$, the first neural collapse metric $NC_1$ is then given by

$$NC_1 = \frac{1}{C} \operatorname{trace}\big(\Sigma_W \Sigma_B^\dagger\big).$$

**NC2:** Let $W \in \mathbb{R}^{C \times p}$ be the classifier weight matrix. The second metric $NC_2$ measures the $\ell_2$ distance between the normalized simplex ETF and the normalized matrix $WW^\top$, i.e.,

$$NC_2 = \left\| \frac{WW^\top}{\|WW^\top\|_F} - \frac{1}{\sqrt{C-1}} \left( I_C - \frac{1}{C} \mathbf{1}_C \mathbf{1}_C^\top \right) \right\|_F.$$

**NC3:** Let the centered class-mean matrix be

$$\dot{M} = [\hat{\mu}_1 \ \hat{\mu}_2 \ \cdots \ \hat{\mu}_C] \in \mathbb{R}^{p \times C}.$$

*Table 1.* Performance comparison of different methods.

| Method | CE | Inv-Freq | Inv-Sqrt | CB | Focal | TCR | IB | Range | Ours |
|--------|-----|----------|----------|-----|-------|-----|-----|-------|------|
| Accuracy(IF=50) | 45.51 | 47.66 | 47.85 | 48.11 | 48.71 | 50.19 | 48.02 | 48.21 | **52.59** |
| Accuracy(IF=100) | 41.62 | 38.19 | 42.45 | 43.34 | 42.13 | 45.35 | 45.14 | 43.03 | **47.88** |
| Accuracy(IF=200) | 36.78 | 25.25 | 35.69 | 39.05 | 38.22 | 40.57 | 40.55 | 38.51 | **43.14** |

The third metric $NC_3$ quantifies the $\ell_2$ distance between the normalized simplex ETF and the normalized matrix $W\dot{M}$:

$$NC_3 = \left\| \frac{W\dot{M}}{\|W\dot{M}\|_F} - \frac{1}{\sqrt{C-1}} \left( I_C - \frac{1}{C}\mathbf{1}_C\mathbf{1}_C^\top \right) \right\|_F .$$

We track the evolution of NC1-NC3 during training on CIFAR-100-LT (Cui et al., 2019b) with the imbalance factor of 100 and compute the NC metrics each epoch. The results of different loss functions are provided in Figure 2. As shown in this figure, we observe that most reweighting methods, including our proposed approach, achieve relatively low NC1 values by the end of training. This is consistent with prior observations in (Dang et al., 2024a) that NC1 is usually the easiest NC property to emerge during training. Compared with the baselines, our method achieves lower NC2 and NC3 values at the end of training, indicating that the learned classifier and class means are closer to the simplex ETF geometry and the self-dual alignment described by Neural Collapse. Moreover, our method also achieves the lowest class-wise imbalance coefficient, further indicating that the proposed inverse reweighting strategy effectively reduces class-wise optimization imbalance.

### 5.3. Comparison with Different Long-Tailed SOTA Methods

In this subsection, we conduct extensive experiments to evaluate our proposed method on widely used long-tailed datasets: CIFAR-10-LT, CIFAR-100-LT (Cui et al., 2019b), iNaturalist (Van Horn et al., 2018), and ImageNet-LT (Deng et al., 2009).

**Baseline** We compare our proposed method with various representative long-tailed approaches. Some existing baselines use contrastive learning: KCL (Kang et al., 2020), TSC (Li et al., 2022), HCL (Wang et al., 2021), B-SCL (Zhao et al., 2026), and FeatRecon (Yi et al., 2025). We also choose some two-stage methods: BBN (Zhou et al., 2020), RIDE (Wang et al., 2020), MisLAS (Zhong et al., 2021), and IP-DPP (Lin & Yuan, 2025). Moreover, CMO (Park et al., 2022), SEL (Jian et al., 2025), and GLMC (Du et al., 2023) apply data augmentation, and FedYoYo (Yan et al., 2025) uses self-supervised learning to improve model performance. There is also a baseline from an optimization

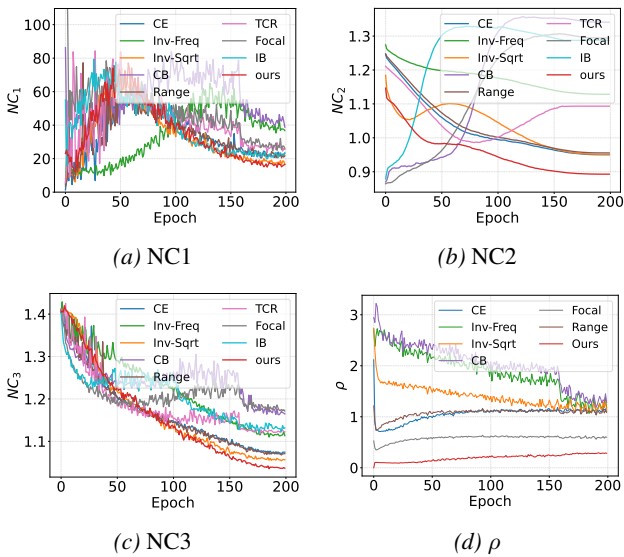

*(a)* NC1  *(b)* NC2

*(c)* NC3  *(d)* $\rho$

*Figure 2.* The evolution of NC1-NC3 metrics and class-wise imbalance coefficients across different loss functions on the CIFAR-100-LT dataset with the imbalance rate of 100.

perspective: Focal-SAM (Li et al., 2025). Meanwhile, some recent methods are motivated by Neural Collapse: INC (Liu et al., 2023), ETF-DR (Yang et al., 2022), and RBL (Peifeng et al., 2023).

**CIFAR-LT** Following the standard evaluation protocol on CIFAR-10-LT and CIFAR-100-LT, we compare our method with other state-of-the-art baselines and report Top-1 accuracy under three imbalance factors (IF=200, 100, 50) in Table 2. In this table, we use two training settings: applying our proposed loss reweighting strategy only and applying the learning-rate schedule together with the proposed loss reweighting method. Our reweighting strategy achieves strong overall improvements across diverse baselines and imbalance ratios. Adding our loss-reweighting method brings extra improvements, showing that the two components work well together and remain effective across long-tailed settings. Under IF=100, the standard cross-entropy baseline obtains 82.0% and 48.9% on CIFAR-10-LT and CIFAR-100-LT, respectively, using the benefits of both components, and gains +6.0% and +7.3% from the previous baseline. When combining with GLMC, our method obtains 84.0%, 89.1%, and 91.1% on CIFAR-10-LT and

*Table 2.* Experiment results on CIFAR-LT. Results marked with * are reproduced by us under the same experimental protocol.

| Method | CIFAR-10-LT | | | CIFAR-100-LT | | |
|---|---|---|---|---|---|---|
| | 200 | 100 | 50 | 200 | 100 | 50 |
| BBN (CVPR20) | / | 79.9 | 82.2 | / | 42.6 | 47.1 |
| KCL (ICLR21) | / | 77.6 | 81.7 | / | 42.8 | 46.3 |
| TSC (CVPR22) | / | 79.7 | 82.9 | / | 43.8 | 47.4 |
| HCL (CVPR21) | / | 81.4 | 85.4 | / | 46.7 | 51.9 |
| MiSLAS (CVPR21) | / | 82.1 | 85.7 | / | 47.0 | 52.3 |
| RIDE-3 experts (ICLR21) | / | 81.6 | 84.0 | / | 48.6 | 51.4 |
| RBL (ICML23) | 81.2 | 84.7 | 87.6 | 48.9 | 53.1 | 57.2 |
| INC-DRW (AISTATS23) | 75.8 | 81.9 | 82.7 | 42.5 | 48.6 | 51.7 |
| DisA (ICML24) | 69.2 | 74.7 | 78.6 | 39.8 | 44.5 | 49.6 |
| IP-DPP (NeurIPS25) | / | 76.4 | / | / | 52.4 | / |
| B-SCL (NeurIPS25) | 84.1 | 88.0 | / | 51.0 | 56.4 | / |
| FeatRecon (ICLR25) | / | 85.2 | 87.8 | / | 52.5 | 57.0 |
| SEL (ICCV25) | / | 75.8 | 79.8 | / | 44.5 | 49.6 |
| FedYoYo (ICCV25) | / | 81.5 | 83.9 | / | 46.1 | 50.8 |
| Focal-SAM (ICML25) | 71.7 | 77.2 | 82.0 | 39.6 | 44.0 | 48.1 |
| CE* | 69.1 | 76.0 | 81.1 | 36.8 | 41.6 | 45.5 |
| CE*+**Loss (Ours)** | 78.6 | 82.3 | 85.7 | 43.1 | 47.9 | 52.6 |
| CE*+**Loss+LR (Ours)** | 76.8 | 82.0 | 85.3 | 43.9 | 48.9 | 52.8 |
| CMO (CVPR22) | 69.6* | 74.8* | 80.0* | 39.0 | 43.9 | 48.3 |
| CMO+**Loss (Ours)** | 76.7 | 82.0 | 85.4 | 43.2 | 46.8 | 52.3 |
| CMO+**Loss+LR (Ours)** | 77.2 | 81.6 | 85.3 | 43.8 | 48.8 | 52.7 |
| ETF-DR (NeurIPS22) | 71.9 | 76.5 | 81.0 | 40.9 | 45.3 | 50.4 |
| ETF-DR+DisA | 73.7 | 78.5 | 81.4 | 41.5 | 45.9 | 51.0 |
| ETF-DR+**Loss (Ours)** | 72.5 | 77.9 | 81.4 | 42.2 | 47.0 | 51.4 |
| ETF-DR+**Loss+LR (Ours)** | 72.3 | 77.9 | 81.8 | 43.4 | 48.0 | 52.9 |
| GLMC (CVPR23) | 83.4* | 87.8 | 90.2 | 50.8* | 55.9 | 61.1 |
| GLMC+SEL | / | 85.4 | 88.6 | / | 56.5 | 58.9 |
| GLMC+**Loss (Ours)** | 83.8 | 88.6 | 90.8 | 52.0 | 58.3 | 63.6 |
| GLMC+**Loss+LR (Ours)** | 84.0 | 89.1 | 91.1 | 52.4 | 58.6 | 63.9 |

*Table 3.* Experiment results on ImageNet-LT. Results marked with * are reproduced by us under the same experimental protocol.

| Method | ImageNet-LT | | | |
|---|---|---|---|---|
| | Many | Med | Few | All |
| KCL (ICLR21) | 61.8 | 49.4 | 30.9 | 51.5 |
| TSC (CVPR22) | 63.5 | 49.7 | 30.4 | 52.4 |
| MiSLAS (CVPR21) | 61.7 | 51.3 | 35.8 | 52.7 |
| RIDE-3 experts (ICLR21) | 66.2 | 51.7 | 34.9 | 54.9 |
| RBL (ICML23) | 64.8 | 49.6 | 34.2 | 53.3 |
| INC-DRW (AISTATS23) | 67.1 | 49.7 | 29.0 | 53.9 |
| DisA (ICML24) | 67.7 | 38.6 | 7.3 | 44.8 |
| IP-DPP (NeurIPS25) | 59.7 | 50.8 | 32.4 | 51.7 |
| FeatRecon (ICLR25) | / | / | / | 56.8 |
| SEL (ICCV25) | 64.1 | 38.4 | 31.3 | 47.9 |
| FedYoYo (ICCV25) | 42.1 | 25.8 | 15.4 | 31.4 |
| Focal-SAM (ICML25) | 63.9 | 52.2 | 34.4 | 54.3 |
| CE* | 67.0 | 36.9 | 4.6 | 44.1 |
| CE*+**Loss (Ours)** | 66.8 | 40.7 | 8.3 | 46.4 |
| CE*+**Loss+LR (Ours)** | 66.4 | 41.0 | 8.8 | 46.5 |
| CMO (CVPR22) | 67.0 | 42.3 | 20.5 | 49.1 |
| CMO+**Loss (Ours)** | 67.4 | 44.1 | 22.1 | 50.3 |
| CMO+**Loss+LR (Ours)** | 65.8 | 44.6 | 22.8 | 49.8 |
| ETF-DR (NeurIPS23) | / | / | / | 44.7 |
| ETF-DR+DisA | 65.2 | 39.9 | 12.8 | 45.3 |
| ETF-DR+**Loss (Ours)** | 63.8 | 40.5 | 13.6 | 45.8 |
| ETF-DR+**Loss+LR (Ours)** | 63.6 | 42.1 | 14.4 | 46.6 |
| GLMC (CVPR23) | 70.1 | 52.4 | 30.4 | 56.3 |
| GLMC+SEL | 68.7 | 54.4 | 38.3 | 57.2 |
| GLMC+**Loss (Ours)** | 67.2 | 55.6 | 38.8 | 57.6 |
| GLMC+**Loss+LR (Ours)** | 67.0 | 55.9 | 39.0 | 57.6 |

52.4%, 58.6%, and 63.9% on CIFAR-100-LT, achieving the SOTA performance.

**ImageNet-LT and iNaturalist** We further evaluate our proposed method on large-scale datasets, ImageNet-LT (Deng et al., 2009) and iNaturalist (Van Horn et al., 2018), and report the results in Table 3 and Table 4. We evaluate two settings: using the proposed loss reweighting strategy alone and using it together with the proposed learning-rate schedule. On the standard cross-entropy baseline, the accuracy reaches 46.5% on ImageNet-LT and 65.8% on iNaturalist by using both components. When applied to the strong baseline GLMC, the accuracy becomes 57.6% on ImageNet-LT and 74.8% on iNaturalist with both the learning rate schedule and the proposed reweighting method. Overall, our method achieves consistent gains across different baselines and attains the SOTA performance on both datasets.

### 5.4. Mechanism Analysis

In this subsection, we further analyze the proposed inverse-view reweighting strategy in long-tailed learning.

**t-SNE Visualization** Guided by the symmetric simplex ETF geometry, we set equal per-class average loss as an ideal objective and solve reweighting from an inverse per-

spective. This yields a closed-form per-class solution that balances the training loss between head and tail classes and brings the training dynamics closer to Neural Collapse (NC). As a result, the NC-consistent geometry leads to noticeably more compact within-class clusters and clearer inter-class separation. As shown in Figure 3, we can see that our method yields tighter clusters and reduced overlap between head and tail classes compared with the cross-entropy baseline. This evidence directly supports our goal of reducing class imbalance by enforcing equal per-class average loss, approaching NC geometry under long-tailed imbalance.

**Ablation** Table 5 further validates the roles of our two reweighting components. We conduct an ablation experiment on CIFAR-100-LT with the imbalance factor of 100. The batch-wise inverse reweighting yields 46.7%, indicating that the batch-level closed-form reweighting effectively balances per-class loss within each mini-batch. Moreover, using only the macro-level reweighting also improves performance to 45.4%, highlighting the value of reducing the class imbalance that builds during training. Combining batch-wise inverse reweighting and macro-level reweighting achieves 47.9%, showing that the two components are complementary.

## 6. Related Work

**Long-Tailed Learning.** In real-world applications, data often follow a long-tailed distribution, where a few head

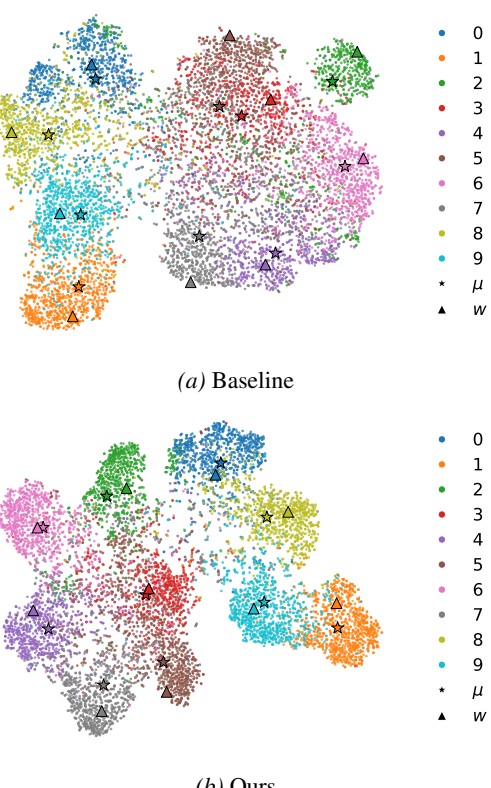

*(a)* Baseline

*(b)* Ours

*Figure 3.* t-SNE visualizations of the cross-entropy baseline and our proposed reweighting on the CIFAR-10-LT with the imbalance factor of 50. Different dots denote the features, ★ and ▲ express the class means and classifier weights for each class, respectively.

*Table 4.* Experiment results on iNaturalist. Results marked with * are reproduced by us under the same experimental protocol.

| Method | iNaturalist 2018 |
| --- | --- |
| | **Accuracy** |
| BBN (CVPR20) | 66.3 |
| KCL (ICLR21) | 68.6 |
| TSC (CVPR22) | 69.7 |
| MiSLAS (AISTATS23) | 71.6 |
| RIDE-3 experts (ICLR21) | 72.2 |
| IP-DPP (NeurIPS25) | 74.0 |
| FeatRecon (ICLR25) | 72.9 |
| SEL (ICCV25) | 69.1 |
| Focal-SAM (ICML25) | 71.8 |
| CE* | 63.5 |
| CE*+**Loss (Ours)** | 66.3 |
| CE*+**Loss+LR (Ours)** | 65.8 |
| CMO (CVPR22) | 64.5* |
| CMO+**Loss (Ours)** | 65.6 |
| CMO+**Loss+LR (Ours)** | 65.1 |
| GLMC (CVPR23) | 72.2 |
| GLMC+SEL | 74.5 |
| GLMC+**Loss (Ours)** | 74.9 |
| GLMC+**Loss+LR (Ours)** | 74.8 |

classes contain abundant samples while most tail classes are severely underrepresented. Long-tailed learning aims to achieve good performance on both head and tail classes. Existing methods can be categorized into three groups: class rebalancing, data augmentation, and model-level improvements. Early approaches adopt oversampling of tail classes (Han et al., 2005) or under-sampling of head classes (Drummond et al., 2003), as well as various reweighting schemes that explicitly adjust class-wise (Cui et al., 2019a) or instance-wise (Lin et al., 2017) loss weights to prevent head classes from dominating training. And (Du et al., 2023) (Vigneswaran et al., 2021) employ stronger data augmentation strategies or design generative models to synthesize new samples. Some model-level methods further improve training by decoupling learning (Kang et al., 2019), calibrating logits (Menon et al., 2020), or classifier weights (Alshammari et al., 2022).

Focusing on reweighting-based approaches, some existing reweighting methods tackle long-tailed recognition by adjusting class-wise or instance-wise loss weights to address the long-tailed distribution. Some methods use class frequency for reweighting like Inv-Freq and Inv-Sqrt apply

the inverse of the sample number, and CB Loss (Cui et al., 2019a) computes the weights by the effective number of samples. And there are also some instance-wise reweighting methods that depend on the sample's influence, such as Focal Loss (Lin et al., 2017) and IB Loss (Park et al., 2021). In addition, Range Loss (Zhang et al., 2017) rebalances instances in the feature space, and two-component reweighting (TCR) (Jamal et al., 2020) applies meta-learning to rebalance long-tailed settings. Beyond frequency or heuristic-based designs, recent studies (He, 2024; Holtz et al., 2022; Heydari et al., 2019) also explore learning-based or dynamic weighting rules that adapt weights according to training dynamics or optimization signals, such as gradient-driven reweighting and bi-level sample reweighting, as well as general adaptive loss-weighting schemes for multi-part objectives.

**Neural Collapse** Neural Collapse (NC) (Papyan et al., 2020) is a geometric phenomenon observed in standard balanced classification problems at the final stage of training, which is commonly interpreted as an idealized training objective for the learning problem. Under NC, the class means approximately form a simplex Equiangular Tight Frame (ETF) on a hypersphere, and the classifier weight vectors align with the corresponding class means, leading to an ideal highly symmetric feature-classifier configuration (NC1–NC4). Recent theoretical studies further characterize how NC geometry changes under class imbalance in unconstrained feature models with cross-entropy loss (Hong & Ling, 2023; Dang et al., 2024b), revealing skewed class-mean structures and imbalance-dependent scaling behaviors. Meanwhile, alternative imbalanced NC structures have also been investigated, such as neural collapse to multiple centers (Yan et al., 2024). Inspired by NC, (Yang et al., 2022) fixes the

*Table 5.* Ablation study

| Setting | Reweighting | | CIFAR-100-LT |
|---|---|---|---|
| | Batch-wise | Macro-level | IF=100 |
| CE (w/o reweighting) | | | 41.6 |
| CE + Batch-wise inverse | ✓ | | 46.7 |
| CE + Macro-level reweighting | | ✓ | 45.4 |
| CE + Batch-wise + Macro-level | ✓ | ✓ | 47.9 |

last-layer classifier as a simplex ETF and proposes the Dot-Regression (ETF-DR) loss and the Representation-Balanced Learning (RBL) framework (Peifeng et al., 2023) further introduces orthogonal matrices to register the sample features and balanced features while the ETF geometric structure is preserved. In addition, (Xie et al., 2023) proposes the Attraction-Repulsion-Balanced (ARB) Loss to balance the gradients among components from different classes. (Gao et al., 2024) proposed Distribution Alignments Optimization (DisA), an OT-based regularizer that matches the distribution of last-layer features to an ideal balanced ETF distribution. (Wang et al., 2026) analyzes the consequence of space misalignment of NC through large deviation theory.

## 7. Conclusion

In this paper, we revisit loss reweighting methods for long-tailed learning and propose an explicit class-wise target of equal per-class average loss, motivated by Neural Collapse (NC). Based on this objective, we formulate reweighting as an inverse problem and derive a closed-form per-class solution for loss reweighting. Extensive experiments show that our method exhibits strong generalization performance while better recovering the ideal NC geometry.

## Impact Statement

This paper presents work whose goal is to advance the field of Machine Learning. There are many potential societal consequences of our work, none of which we feel must be specifically highlighted here.

## Acknowledgments

The work was partially supported by the following: the Zhejiang Provincial Natural Science Foundation - Exploration Project under No. LMS26F020007, the Wenzhou Applied Fundamental Research Program (Basic Research) under No. GG20250198, the WKU 2026 International Frontier Interdisciplinary Research Institute Talent Program under No. WKUTP2026002, the WKU 2025 International Collaborative Research Program under No. ICRPSP2025001.

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

# A. Reweighted Loss Methods

In this section, we summarize commonly used loss reweighting methods in long-tailed classification. We present the class weighting used by each method and provide the corresponding loss functions.

## A.1. Frequency-based Reweighting

A common and simple reweighting strategy assigns each class a loss weight inversely proportional to its sample numbers. Let $n_c$ denote the number of samples in class $c$.

**Inverse-Frequency (Inv-Freq).** The Inv-Freq weighting sets

$$w_c^{\text{Inv-Freq}} = \frac{1}{n_c}.$$

Using this class weight, the corresponding reweighted loss over the training set is

$$L(W) = \frac{1}{N} \sum_{c=1}^{C} \sum_{i=1}^{n_c} w_c^{\text{Inv-Freq}} \ell\big(W, h(x_{i,c}), c\big).$$

This heuristic increases the contribution of minority classes while reducing the influence of majority classes.

**Inverse-Square-Root (Inv-Sqrt).** To soften this effect, a smoother variant replaces the inverse with the inverse square root:

$$w_c^{\text{Inv-Sqrt}} = \frac{1}{\sqrt{n_c}}.$$

The corresponding reweighted loss becomes

$$L(W) = \frac{1}{N} \sum_{c=1}^{C} \sum_{i=1}^{n_c} w_c^{\text{Inv-Sqrt}} \ell\big(W, h(x_{i,c}), c\big).$$

where $N = \sum_{c=1}^{C} n_c$ is the total number of training samples. In contrast to Inv-Freq, Inv-Sqrt achieves a more moderate form of rebalancing while avoiding excessively large weights for rare classes in long-tailed classification.

## A.2. Class-Balanced Loss

The Class-Balanced (CB) loss (Cui et al., 2019a) replaces the raw class frequency $n_c$ by using its effective number of samples. For a class $c$ with $n_c$ training examples, the effective number is defined as

$$E_{n_c} = \frac{1 - \beta^{n_c}}{1 - \beta},$$

where $\beta \in [0, 1)$ is a hyper-parameter controlling how aggressively head classes are down-weighted. The corresponding class weight is taken as the inverse effective number,

$$w_c^{\text{CB}} = \frac{1}{E_{n_c}} = \frac{1 - \beta}{1 - \beta^{n_c}}.$$

Using these weights in our generic reweighted loss function, the resulting Class-Balanced loss is

$$L(W) = \frac{1}{N} \sum_{c=1}^{C} \sum_{i=1}^{n_c} w_c^{\text{CB}} \ell\big(W, h(x_{i,c}), c\big),$$

where $N = \sum_{c=1}^{C} n_c$ denotes the total number of training samples. When $\beta = 0$, $w_c^{\text{CB}} = 1$ (no reweighting), and as $\beta \to 1$, $w_c^{\text{CB}}$ approaches the inverse-frequency weight $1/n_c$.

## A.3. Focal Loss

The Focal loss (Lin et al., 2017) is designed to handle the severe foreground–background imbalance characteristic of dense prediction problems. While the standard cross-entropy loss assigns a similar magnitude of loss to both easy and hard examples, the Focal loss reshapes the loss so that well-classified examples receive significantly lower weights, encouraging the model to focus more on hard and informative examples.

For a sample $(x_{i,c}, y_{i,c})$ belonging to class $c$, let $p_c(x_{i,c}; W)$ denote the predicted probability of the ground-truth class. The Focal loss introduces a modulating factor $(1 - p_t)^\gamma$ with a focusing parameter $\gamma \geq 0$ to determine how strongly the loss suppresses easy examples:

$$\mathrm{FL}(x_{i,c}, c; W) = -(1 - p_c(x_{i,c}; W))^\gamma \log p_c(x_{i,c}; W).$$

To further account for class imbalance, an additional balancing factor $\alpha_t \in [0, 1]$ may be included, leading to the $\alpha$balanced version:

$$\mathrm{FL}(x_{i,c}, c; W) = -\alpha_c(1 - p_c(x_{i,c}; W))^\gamma \log p_c(x_{i,c}; W).$$

When $\gamma = 0$, the loss reduces to standard cross-entropy. Increasing $\gamma$ strengthens the down-weighting of easy examples, causing the loss to concentrate on hard samples.

Based on the above definition, the overall Focal loss on the training set can be written as

$$L(W) = \frac{1}{N} \sum_{c=1}^{C} \sum_{i=1}^{n_c} \mathrm{FL}(x_{i,c}, c; W),$$

where $N = \sum_{c=1}^{C} n_c$ denotes the total number of training samples.

## A.4. Influence-Balanced Loss

The Influence-Balanced (IB) Loss (Park et al., 2021) aims to down-weight training samples that have a disproportionately large influence on the decision boundary. Following the notation we defined before, a sample $(x_{i,c}, y_{i,c})$ from class $c$, let $h(x_{i,c}) \in \mathbb{R}^p$ denote the last-layer feature, and let

$$p(x_{i,c}; W) = \big(p_1(x_{i,c}; W), \ldots, p_C(x_{i,c}; W)\big)^\top$$

be the softmax probability vector. We denote by $y^{(c)} \in \{0, 1\}^C$ the one-hot vector of class $c$.

Follow (Park et al., 2021), an influence-balanced weight factor for $(x_{i,c}, y_{i,c})$ is defined as

$$\mathrm{IB}(x_{i,c}, c; W) = \big\| p(x_{i,c}; W) - y^{(c)} \big\|_1 \big\| h(x_{i,c}) \big\|_1,$$

and its inverse is used to attenuate highly influential samples. The sample-wise IB loss then takes the form

$$\ell_{\mathrm{IB}}\big(W, h(x_{i,c}), c\big) = \frac{\ell\big(W, h(x_{i,c}), c\big)}{\mathrm{IB}(x_{i,c}, c; W) + \varepsilon},$$

where $\ell(W, h(x_{i,c}), c)$ is the standard cross-entropy loss and $\varepsilon > 0$ is a small constant for numerical stability.

Using this definition, the overall IB Loss over the training set is

$$L(W) = \frac{1}{N} \sum_{c=1}^{C} \sum_{i=1}^{n_c} \ell_{\mathrm{IB}}\big(W, h(x_{i,c}), c\big),$$

where $N = \sum_{c=1}^{C} n_c$ denotes the total number of training samples.

To further compensate for inter-class imbalance, we introduce a class-wise coefficient $\lambda_c$ that is inversely proportional to the number of training samples in class $c$:

$$\lambda_c = \frac{\alpha \, n_c^{-1}}{\sum_{c'=1}^{C} n_{c'}^{-1}},$$

where $\alpha > 0$ is a scaling hyper-parameter. The resulting class-reweighted IB loss is

$$L(W) = \frac{1}{N} \sum_{c=1}^{C} \sum_{i=1}^{n_c} \lambda_c \, \ell_{\mathrm{IB}}\big(W, h(x_{i,c}), c\big).$$

## A.5. Range Loss

Range Loss (Zhang et al., 2017) is designed to simultaneously reduce intra-class variations and enlarge inter-class separations, particularly for long-tailed distributions. For each class in a mini-batch, let $\{h(x_{i,c})\}_{i=1}^{n_c}$ denote its feature vectors defined before.

**Intra-Class Term.** For class $c$, consider all pairwise Euclidean distances among $\{h(x_{i,c})\}_{i=1}^{n_c}$,

$$d_{i,j}^{(c)} = \left\| h(x_{i,c}) - h(x_{j,c}) \right\|_2, \quad 1 \leq i < j \leq n_c.$$

We sort these distances in descending order and denote by $\{D_j^{(c)}\}_{j=1}^k$ the $k$ largest ones. The intra-class range loss of class $c$ is defined as the harmonic mean of these $k$ ranges:

$$L_c^{\text{intra}} = \frac{k}{\sum_{j=1}^{k} \dfrac{1}{D_j^{(c)}}}.$$

Let $\mathcal{I}$ be the set of classes that appear in the current mini-batch. The overall intra-class term is then

$$L_{\text{intra}} = \sum_{c \in \mathcal{I}} L_c^{\text{intra}}.$$

**Inter-Class Term.** Let $m_c$ denotes the feature center of class $c$,

$$m_c = \frac{1}{n_c} \sum_{i=1}^{n_c} h(x_{i,c}),$$

and define $D_{\text{center}}$ as the shortest center-to-center distance among all class pairs:

$$D_{\text{center}} = \min_{c \neq c'} \left\| m_c - m_{c'} \right\|_2.$$

The inter-class range loss encourages class centers to be well separated by enforcing a margin on this minimum distance:

$$L_{\text{inter}} = \max\left( M - D_{\text{center}}, \, 0 \right),$$

where $M > 0$ is a margin hyper-parameter.

**Overall Loss.** Combining both terms yields the Range Loss:

$$L_{\text{Range}} = \alpha L_{\text{intra}} + \beta L_{\text{inter}},$$

where $\alpha$ and $\beta$ control the relative importance of the two components. Following (Zhang et al., 2017), Range Loss is typically used together with the standard softmax cross-entropy:

$$L(W) = L_{\text{CE}}(W) + \lambda L_{\text{Range}},$$

where $\lambda$ is a balancing coefficient.

## A.6. Two-Component Reweighting Loss

Two-component Reweighting (TCR) Loss (Jamal et al., 2020) aims to minimize the expected risk under a target distribution that is more class-balanced than the long-tailed training distribution. By importance weighting, the target error can be written as an expectation under the training distribution with sample-wise weights. TCR decomposes this weight into a *class-wise* component and an *instance-wise* component.

**Class-Wise Reweighting.** For class $c$, let $n_c$ be the number of training samples in this class. Follow the class-balanced loss (Cui et al., 2019a) based on the effective number of samples, a class-wise weight $w_c$ is defined as

$$w_c = \frac{1 - \beta}{1 - \beta^{n_c}}, \qquad \beta \in [0, 1).$$

This term increases the contribution of tail classes (small $n_c$) and down-weights head classes (large $n_c$).

**Instance-Wise Reweighting.** Beyond class frequencies, different instances within the same class may contribute differently to the target distribution. TCR introduces an additional instance-wise weight $\varepsilon_{i,c}$ for each training example. The total weight of this instance is

$$\alpha_{i,c} = w_c + \varepsilon_{i,c}.$$

The instance-wise weights $\varepsilon_{i,c}$ are learned via a meta-learning procedure using a small class-balanced validation set: instances that are more helpful for improving performance on this balanced set receive larger positive $\varepsilon_{i,c}$, while less useful or noisy instances are assigned small weights.

**Overall TCR Loss.** Combining both components, each instance contributes its loss multiplied by $\alpha_{i,c}$. Let $N = \sum_{c=1}^{C} n_c$ be the total number of training instances. The TCR reweighted loss can be written as

$$L_{\text{TCR}}(W) = \frac{1}{N} \sum_{c=1}^{C} \sum_{i=1}^{n_c} (w_c + \varepsilon_{i,c}) \, \ell\big(W, h(x_{i,c}), c\big).$$

When combined with the standard cross-entropy loss

$$L_{\text{CE}}(W) = \frac{1}{C} \sum_{c=1}^{C} L_c(W), \qquad L_c(W) = \frac{1}{n_c} \sum_{i=1}^{n_c} \ell\big(W, h(x_{i,c}), c\big),$$

the overall training objective becomes

$$L(W) = L_{\text{CE}}(W) + \lambda \, L_{\text{TCR}}(W),$$

where $\lambda > 0$ is a balancing coefficient controlling the strength of the reweighting term.

## B. Proofs

### B.1. Proof For Theorem 3.1

*Proof.* Following (Papyan et al., 2020), under NC1-NC3 and optimal linear decoding, the last-layer classifier can be chosen with zero bias(b=0). Hence, we omit the bias term in the following analysis. Let $\ell$ be any label-symmetric per-sample loss that follows the previous notation. For each class $c \in C$, we have $\mu_c = \mu_G + \hat{\mu}_c$, where $\mu_G$ is a constant. Thus, we can equivalently do the derivation in the feature space of decentralized systems. Since NC3, the centered class means are aligned with the classifier weights. For $\alpha > 0$, we have

$$w_k = \alpha \hat{\mu}_k, \quad k = 1, ..., C$$

For $\hat{\mu}_c$, the logits are :

$$z_k = \langle w_k, \hat{\mu}_c \rangle = \alpha \langle \hat{\mu}_k, \hat{\mu}_c \rangle.$$

Since NC2, the centered feature means will form a Simplex ETF. Hence, there exists $r > 0$ that:

$$\langle \hat{\mu}_c, \hat{\mu}_c \rangle = r^2, \qquad \langle \hat{\mu}_k, \hat{\mu}_c \rangle = -\frac{r^2}{C - 1}, \quad \forall k \neq c.$$

Therefore:

$$z_c = \alpha r^2, \qquad z_k = -\alpha \frac{r^2}{C - 1}, \quad k \neq c.$$

Therefore, for every class c, the logit vector z for a (collapsed) sample of class c has the same pattern. Thus, we have:

$$L_c(W) = \frac{1}{n_c} \sum_{i=1}^{n_c} \ell(W, h(x_{i,c}), c)$$

$$= \frac{1}{n_c} \sum_{i=1}^{n_c} \psi(z(x_{i,c}), c) \text{ is independent of c}$$

which implies

$$L_1(W) = L_2(W) = \cdots = L_C(W).$$

This completes the proof. $\qquad\square$

### B.2. Proof For Theorem 3.2

*Proof.*

**Continuity:** For fixed data, each class-wise loss $L_c(W)$ is a continuous function of $W$. Hence, the mean loss $\bar{L}(W) = \frac{1}{C} \sum_{c=1}^{C} L_c(W)$ is continuous in $W$, and the numerator of $\rho(W)$ in Eq. (10) is obtained from $\{L_c(W)\}$ by algebraic operations and a square root, which are continuous on their domain. Therefore, on the set $\{W : \bar{L}(W) > 0\}$, the map $W \mapsto \rho(W)$ is continuous.

**ETF Solutions Lie in the Zero-Imbalance Set:** We define the set of ETF solutions in the neural collapse sense as follows

$$\mathcal{S}_{\text{ETF}} := \{ W : W \text{ satisfies NC1-NC3} \}$$

By Theorem 3.1, every $W \in \mathcal{S}_{\text{ETF}}$ has equal class-wise losses, $L_1(W) = \cdots = L_C(W)$, and thus $\rho(W) = 0$. In particular,

$$\mathcal{S}_{\text{ETF}} \subseteq \{ W : \rho(W) = 0 \}.$$

**Persistent Imbalance Propagates to Limit Points.** Consider any convergent subsequence $\{W_{t_k}\}_{k \geq 1}$ of $\{W_t\}$ with $t_k \to \infty$ and

$$W_{t_k} \xrightarrow[k \to \infty]{} W_\star.$$

By the persistent imbalance condition where $\rho(W_t) \geq \varepsilon$. For all sufficiently large $k$ we have $t_k \geq T$ and therefore

$$\rho(W_{t_k}) \geq \varepsilon.$$

By continuity of $\rho(\cdot)$

$$\rho(W_\star) = \lim_{k \to \infty} \rho(W_{t_k}) \geq \varepsilon > 0.$$

Hence, every limit point $W_\star$ of the training dynamics lies in the set

$$\{ W : \rho(W) \geq \varepsilon \}.$$

**Separation from the ETF Solution Set.** Combining Step 2 and Step 3, we see that ETF-solutions lie in the set $\{W : \rho(W) = 0\}$, while limit points under the imbalance loss assumption lie in the set $\{W : \rho(W) \geq \varepsilon\}$ with $\varepsilon > 0$. These two sets are disjoint, so no limit point $W_\star$ of $\{W_t\}$ can belong to $\mathcal{S}_{\text{ETF}}$. $\qquad\square$

### B.3. Proof For Theorem 4.1

*Proof.* Denote $L_c = L_c(W)$ and $\bar{L} = \bar{L}(W)$ and define

$$\phi_c(w_c) := \left(w_c L_c - \bar{L}\right)^2 + \alpha\left(w_c - w_c^{(0)}\right)^2.$$

This is a quadratic function in $w_c$ with a leading coefficient $L_c^2 + \alpha > 0$, hence $\phi_c$ is strictly convex and has a unique global minimizer characterized by the first-order optimality condition $\frac{d}{dw_c}\phi_c(w_c) = 0$.

Differentiating and setting the derivative to zero gives

$$2\big(w_c L_c - \bar{L}\big) L_c + 2\alpha\big(w_c - w_c^{(0)}\big) = 0,$$

which simplifies to

$$(L_c^2 + \alpha)\, w_c = \bar{L}\, L_c + \alpha\, w_c^{(0)}.$$

Solving for $w_c$ yields

$$w_c^\star = \frac{\bar{L}\, L_c + \alpha\, w_c^{(0)}}{L_c^2 + \alpha},$$

which is exactly (13). Therefore, $w_c^\star(W)$ is the unique minimizer of (12), completing the proof. $\qquad\square$

## C. MiLe-LR: Mittag–Leffler Learning-Rate Scheduling for Long-Tailed Learning

In long-tailed learning, head classes provide dense and consistent gradients while tail classes provide sparse gradient signals. Thus, long-tailed learning is actually a multi-timescale problem where tail classes are typically fitted at a later stage of training. Since the learning rate controls the training dynamics, we thus seek a schedule that decays reasonably fast while retaining non-negligible late-stage updates to preserve a long effective plasticity horizon for tail classes. Thus, we parameterize the learning rate decay via the Mittag-Leffler (ML) function and introduce a two-stage switch to align with the multi-timescale dynamic of long-tailed learning.

**Mittage-Leffer Function**    The two-parameter ML function is

$$E_{a,b}(x) = \sum_{k=0}^{\infty} \frac{x^k}{\Gamma(ak+b)}, \qquad a > 0,\ b > 0. \tag{16}$$

and we use the one-parameter case $(b = 1)$:

$$E_a(x) \triangleq E_{a,1}(x) = \sum_{k=0}^{\infty} \frac{x^k}{\Gamma(ak+1)}. \tag{17}$$

A key property on the negative real axis is the asymptotic form (for $z > 0$ and large $z$)

$$E_a(-z) \approx \frac{1}{z\,\Gamma(1-a)}, \tag{18}$$

which decays more slowly than exponential schedules. These slow decay intermittent tail-class gradients can still induce meaningful parameter updates.

**Imbalance-Adaptive Tail Strength.**    We adapt the tail strength via $a$ using the normalized entropy of class counts $\{n_c\}_{c=1}^C$. Let $p_c = n_c / \sum_j n_j$ and

$$H_{\mathrm{norm}} = \frac{-\sum_{c=1}^C p_c \log p_c}{\log C}, \tag{19}$$

then we set

$$a = 0.25 + 0.75\, H_{\mathrm{norm}}. \tag{20}$$

More severe imbalance yields lower entropy and thus smaller $\alpha$, producing heavier tails and stronger late-stage driving for tail fitting.

**Scheduler Definition.**  MiLeLR is an *iteration-level* scheduler. Let $\eta_0$ be the base LR and $\eta_t$ the LR at global iteration $t$. We can apply an optional linear warm-up mechanism for the first $T_w$ iterations:

$$\eta_t \; = \; \eta_0 \cdot \frac{t+1}{T_w}, \qquad t < T_w. \tag{21}$$

After warm-up, define the post-warmup index $\tau = t - T_w$, the post-warmup horizon $T = T_{\text{all}} - T_w$, and the switch point $T_s$ (in iterations). We set

$$\eta_t \; = \; \eta_0 \cdot f(\tau), \tag{22}$$

where $f(\tau)$ is defined by the following two stages.

**Stage I (Early Stabilization).**  For $\tau < T_s$, we keep the ML argument in the series-stable regime:

$$p = \frac{\tau}{\max(T_s, 1)}, \qquad z_1 = (1-\varepsilon)p, \qquad f(\tau) = E_\alpha(-z_1). \tag{23}$$

**Stage II (Late Power-Law Driving).**  For $\tau \geq T_s$, we enter the power-law regime to preserve non-vanishing late updates:

$$\tau_2 = \tau - T_s, \qquad T_2 = \max(T - T_s, 1), \qquad s_2 = \min\left(\frac{\tau_2}{T_2}, 1-\varepsilon\right), \tag{24}$$

$$z_2 = 1 + \frac{s_2}{1 - s_2 + \varepsilon}, \qquad f(\tau) \approx \frac{1}{z_2 \, \Gamma(1-\alpha)}. \tag{25}$$

In practice, we compute $E_\alpha(-z)$ with a stable piecewise approximation: a truncated series for $z < 1$ and the asymptotic tail in Eq. (18) for $z \geq 1$, which also motivates the explicit tail form used in Stage II.

**Switch Steps in Practice.**  Let `iters_per_epoch` denote the number of iterations per epoch. We set $T_{\text{all}} = \text{epochs} \times$ `iters_per_epoch` and $T_w = \text{warmup\_epochs} \times$ `iters_per_epoch`. Given a `lr_switch_epoch`, we convert it into iterations and subtract the warm-up to obtain

$$T_s \; = \; \max\left(\lfloor \text{lr\_switch\_epoch} \cdot \text{iters\_per\_epoch} \rfloor - T_w, \, 0\right). \tag{26}$$

# D. Algorithm

The complete algorithm is shown in the Algorithm 1.

# E. Sensitive Analysis

To analyze the sensitivity of hyper-parameters $\alpha$ and $\gamma$ of our proposed method, we conduct a sensitivity analysis on CIFAR-100-LT with the imbalance factor of 100. We report the result with different $\alpha$ and $\gamma$ in Figure 4. As shown in the left panel of Figure 4, we can see that increasing $\gamma$ consistently improves the overall Top-1 accuracy. Here, $\gamma$ controls the strength of our macro-level compensation, which up-weights classes that appear less in mini-batches. This indicates that the macro-level compensation can effectively improve performance. We further study the right panel of Figure 4, where we fix $\gamma = 1$ and vary $\alpha$. In our experiments, the prior is set to a uniform weighting ($w^{(0)} = 1$). We observe that $\alpha = 0$ achieves the best accuracy, and the performance decreases as $\alpha$ increases. This suggests that the uniform prior may not always be beneficial, and a weaker or no prior could gain better results in this setting.

# F. Experiment Details

We perform our experiments on CIFAR-10-LT, CIFAR-100-LT, iNaturalist, and ImageNet-LT. All experiments are implemented by the SGD optimizer with a momentum of 0.9 and weight decay of 5e-4. We set the batch size as 256 for all datasets. We use a multi-stage learning rate schedule, and the initial learning rate is 0.1. For CIFAR-LT, we use ResNet-32 as the backbone and train for 200 epochs. For iNaturalist and ImageNet-LT, we use ResNet-50 as the backbone and train for 90 epochs.

---

**Algorithm 1** Batch-wise Inverse Reweighting with Macro-level Compensation (Closed-form)

---

**Input:** Training set $\mathcal{D}$; base loss $\ell_{\text{base}}$; prior class weights $\{w_c^{(0)}\}_{c=1}^C$; regularization coefficient $\alpha$; macro compensation
       exponent $\gamma \geq 0$; network parameters $W$

Initialize batch-appearance counters $B_c \leftarrow 0$ for all classes $c = 1, \ldots, C$

**for** *each epoch* **do**
    **for** *each mini-batch* $\mathcal{B} = \{(x_i, y_i)\}_{i=1}^m \subset \mathcal{D}$ **do**
        `/* Compute base loss for samples in the batch                                    */`
        $\ell_i \leftarrow \ell_{\text{base}}(f(x_i; W), y_i), \quad i = 1, \ldots, m$
        `/* Classes appearing in the batch                                               */`
        $\mathcal{I}_\mathcal{B} \leftarrow \{c \mid \exists i, \, y_i = c\}$
        `/* (A) Update macro-level batch-frequency statistics                            */`
        **foreach** $c \in \mathcal{I}_\mathcal{B}$ **do**
          $B_c \leftarrow B_c + 1$
        `/* (B) Estimate per-class mean loss in the batch                                */`
        **foreach** $c \in \mathcal{I}_\mathcal{B}$ **do**
          $I_c \leftarrow \{i \mid y_i = c\} \quad \hat{L}_c \leftarrow \frac{1}{|I_c|} \sum_{i \in I_c} \ell_i$
        `/* Batch average of class-mean losses                                           */`
        $\hat{L}_{\text{avg}} \leftarrow \frac{1}{|\mathcal{I}_\mathcal{B}|} \sum_{c \in \mathcal{I}_\mathcal{B}} \hat{L}_c$
        `/* (C) Closed-form batch-wise inverse reweighting (consistent with Eq.(13))`
        `   */`
        **foreach** $c \in \mathcal{I}_\mathcal{B}$ **do**
          $w_c^\star \leftarrow \dfrac{\hat{L}_{\text{avg}}\hat{L}_c + \alpha \, w_c^{(0)}}{\hat{L}_c^2 + \alpha}$
        `/* (D) Macro-level compensation (consistent with Eq.(14)--(15))                 */`
        **foreach** $c \in \mathcal{I}_\mathcal{B}$ **do**
          $\beta_c \leftarrow B_c^{-\gamma}$
        $\bar{\beta} \leftarrow \frac{1}{|\mathcal{I}_\mathcal{B}|} \sum_{c \in \mathcal{I}_\mathcal{B}} \beta_c$
        **foreach** $c \in \mathcal{I}_\mathcal{B}$ **do**
          $\hat{w}_c \leftarrow w_c^\star \cdot \dfrac{\beta_c}{\bar{\beta}}$
        `/* Weighted loss and update                                                     */`
        $\mathcal{L}_\mathcal{B} \leftarrow \frac{1}{m} \sum_{i=1}^m \hat{w}_{y_i} \ell_i$
        Update $W$ using $\nabla_W \mathcal{L}_\mathcal{B}$

---

For the reweighting component, we adopt a two-stage switching strategy controlled by a switch epoch. We train with the original loss before the switch epoch and apply our reweighting afterwards. For the standard cross-entropy baseline, we keep the hyper-parameter $\alpha$ as 0 and the strength of the macro compensation $\gamma$ as 1.0 under all three different imbalance factors on all datasets. We set $\alpha$ as 0.1 and $\gamma$ as 1.0 when using ETF-DR as a baseline on all datasets. For CMO, the hyper-parameter $\alpha$ and $\gamma$ are 0 and 0.5 on all datasets. We apply our reweighting on GLMC with $\alpha$ as 0 and $\gamma$ as 1 on all datasets.

For the proposed learning-rate schedule, we use a learning-rate switch epoch to switch the mode. We set the switch point at epoch 160 on CIFAR, epoch 60 on iNaturalist, and ImageNet. For the standard cross-entropy baseline, the initial learning rate is 0.45, 0.35 and 0.4 with the imbalance factor of 50, 100, 200 on CIFAR-100-LT and 0.35, 0.3, and 0.3 with the imbalance factor of 50, 100, 200 on CIFAR-10-LT. We set the initial learning rate as 0.35 on iNaturalist and ImageNet for the cross-entropy baseline. For the ETF-DR baseline, the initial learning rate is 0.15, 0.25, and 0.2 on CIFAR-10-LT and 0.5, 0.55, and 0.5 on CIFAR-100-LT with the imbalance factor of 50, 100, and 200, respectively. For the CMO baseline, we apply our learning rate schedule by using the initial learning rate as 0.25, 0.2, and 0.15 on CIFAR-10-LT and 0.45, 0.2, and 0.35 on CIFAR-100-LT under the imbalance factor IF=50,100,200. We set the initial learning rate as 0.35 on iNaturalist and ImageNet for the CMO baseline. For GLMC, we set the initial learning rate to 0.15.

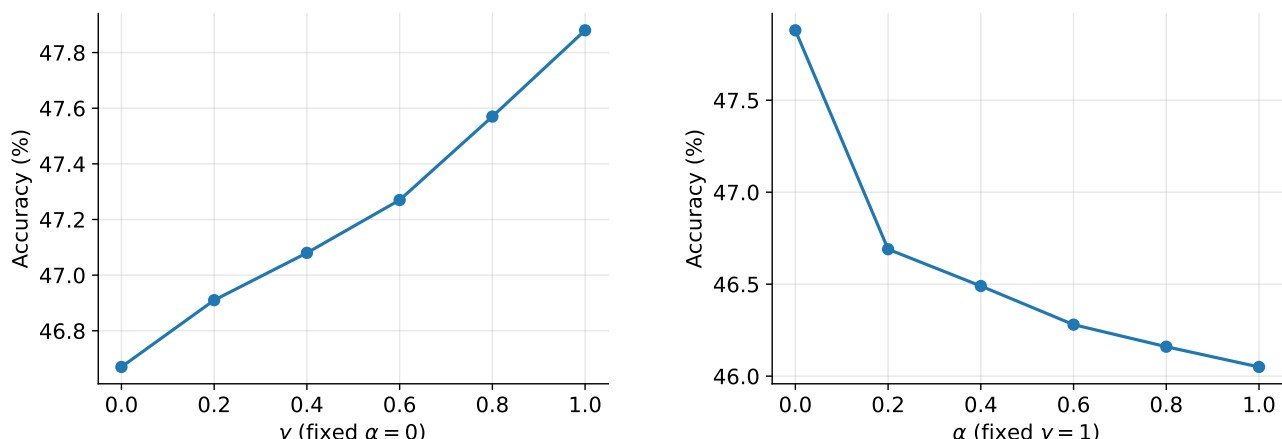

*Figure 4.* Sensitive analysis of hyper-parameters. (Left) Different $\gamma$ with fixed $\alpha = 0$ to control the macro-level compensation across mini-batches. (Right) Different $\alpha$ with fixed $\gamma = 1$ to control the strength of the Tikhonov regularization towards the prior weights $w^{(0)}$ in the inverse reweighting.

## G. Computational Cost Discussion

Our method introduces only small runtime and memory overhead, enabled by a closed-form per-class solution for the reweighting coefficients. In each iteration, we directly compute the class-wise weights by a closed-form per-class solution from a simple batch-level class statistics, without training an additional weighting network or performing extra backpropagation for weight updates. Consequently, the additional cost mainly comes from lightweight per-class reductions and vector operations to obtain each class-weighting parameter. Compared with the standard forward and backward pass, this extra computation is typically limited and does not significantly affect training efficiency.

