# OpenReview forum: "Rethinking Loss Reweighting for Imbalance Learning as an Inverse Problem: A Neural Collapse Point of View"
_ICML.cc/2026/Conference — ICML 2026 regular_

### Official Review · Reviewer_jcbC · 2026-03-07

**Soundness:** 3
**Presentation:** 3
**Significance:** 2
**Originality:** 3
**Overall Recommendation:** 4
**Confidence:** 4

**Summary:**

The paper proposes a reweighting method for the long-tailed classification problem, inspired by the Neural Collapse phenomenon. The strategy is to reweight samples during training so that the per-class average loss becomes equal. The experiments show that the proposed reweighting method performs better than other reweighting strategies in the literature.

**Compliance With Llm Reviewing Policy:**

Affirmed.

**Final Justification:**

My previous concerns are mostly resolved, and I have increased my score from 3 to 4. With the additional experiments comparing the evolution of the three NC metrics (similar to Figure 2) on methods that regularize the weight/feature structure toward balanced ETF geometry (e.g., the INC method [1]), as well as experiments comparing performance with these methods, the paper provides a clearer picture of the relationship between NC structure and generalization performance and better supports the proposed method.

[1] Liu, X., Zhang, J., Hu, T., Cao, H., Yao, Y., and Pan, L. Inducing neural collapse in deep long-tailed learning. AISTATS 2023

**Key Questions For Authors:**

From Theorem 3.1, we see that under the ETF convergence structure, every class achieves the same class-wise average loss. My question is about the converse: does driving the class-wise average losses to be equal induce the desired ETF structure? A broader question is how we know that a perfectly balanced ETF is beneficial for long-tailed problems and why it is chosen as the target here.

I am happy to update my score if my concerns are addressed or clarified in the rebuttal.

**Limitations:**

Limitations are not discussed.

**Strengths And Weaknesses:**

**Strengths:**
- The paper is well-presented and easy to follow, code is available.
- The proposed reweighting method is simple and novel.
- The experiments in Table 1 show that the proposed method performs better than other reweighting strategies in the literature.

**Weaknesses:**
- I understand the goal of achieving equal per-class average loss inspired by the balanced ETF structure. However, the NC2 and NC3 metrics remain quite large and are still far from the ETF. This may be due to the large CIFAR100 dataset with a high imbalance ratio. I wonder how the results would look for a smaller dataset like CIFAR10, with a lower imbalance factor, say 10.

- In the experiments with other SOTA methods, it seems that GLMC + SEL already provides a strong baseline, so the improvement from the proposed method on top of it is not very significant.

- The paper lacks a discussion of limitations and future research directions.

---

> ### Author Rebuttal · Authors · 2026-03-31
>
> We thank the reviewer for these comments.
>
> **W1: Additional NC metrics results on an easier setting.**
>
> We agree that the NC2 and NC3 values on CIFAR-100-LT with IF=100 are still not very small in an absolute sense. To clarify this point, we additionally evaluate NC2 and NC3 metrics (see nc\_metrics.png) on CIFAR-10-LT and CIFAR-100-LT under different imbalance factors (IF=20, 50, 100, 200). As reported in our response to Reviewer YRWX (KQ5), on CIFAR-10-LT and under smaller imbalance factors, NC2 and NC3 are consistently lower, and they further decrease as the imbalance factor becomes smaller. This confirms that easier and more balanced settings indeed exhibit stronger alignment with NC geometry. In our paper, our claim is not that long-tailed training should fully recover the balanced data of the simplex ETF geometry. As stated in Theorem 3.2, persistent class-wise loss imbalance precludes convergence to the ideal ETF solution. Therefore, in the imbalanced long-tailed settings, a more realistic expectation is improved approximation to NC rather than exact recovery. Empirically, compared with other reweighting methods, it yields the lowest NC metrics and loss-imbalance coefficient, while also achieving the best accuracy. We have provided the details in our manuscript. Hence, even if exact NC is not fully reached, moving closer to NC is already meaningful and beneficial.
>
> Results are shown in https://anonymous.4open.science/r/anonymous-repo-A24D
>
> **W2: GLMC+SEL provides a strong baseline.**
>
> We agree that GLMC+SEL is a strong baseline. However, the gain of our method on top of GLMC is not marginal. In fact, on the shared settings, our method consistently outperforms GLMC+SEL across datasets, including CIFAR-LT, ImageNet-LT, and iNaturalist, and achieves SOTA performance in several of these settings. Specifically, on CIFAR-100-LT, our approach achieves gains of +2.1 and +5.0 points when plugged into GLMC over GLMC+SEL baseline at imbalance factors 100 and 50, respectively. We also note that GLMC+SEL is not consistently stronger than GLMC itself on CIFAR, whereas our method provides a more stable improvement when plugged into GLMC.
>
> **W3: Lack of limitation and future work.**
>
> We do not believe this affects the technical validity or contribution of the paper, since the paper already makes the scope of the method clear, including its focus on long-tailed classification and the assumptions underlying the proposed formulation. If necessary, we are happy to add a brief paragraph on limitations and future directions in the revision, including the current focus on standard long-tailed classification settings and possible extensions to broader imbalance-related scenarios.
>
> **KQ1: Whether equal-loss balancing induces ETF, why ETF is beneficial for long-tailed learning, and why it is chosen as the target.**
>
> In our paper, equal per-class average loss is used as a necessary property of the desired NC-consistent ETF end state. Moreover, Theorem 3.2 shows that persistent loss imbalance precludes convergence to ETF. Therefore, driving class-wise average losses toward equality is not used as a sufficient condition for exact ETF recovery, but rather as a principled reweighting target. In particular, our method achieves smaller NC metrics compared with other baseline methods, shown in Figure 2. In addition, our method also brings gains when plugged into ETF-DR, which already fixes the classifier as a simplex ETF. These results suggest that, although equal-loss balancing alone does not fully determine ETF, it is an effective target for moving long-tailed training toward a more NC-consistent regime.
>
> Moreover, as we clarify in our response to Reviewer pXv5 (KQ2), the simplex ETF is beneficial for long-tailed learning because it provides a more desirable class geometry and is optimal in the large-deviation error exponent sense. As for why it is chosen as the target here, the key point is that our method rethinks loss reweighting itself. Since reweighting operates on class contributions in the loss rather than directly imposing geometric constraints on features or classifiers, the target should be an explicit and optimizable quantity in the loss space. In this sense, ETF serves as the geometric end-state reference, while equal per-class average loss is the corresponding loss-level property. This is why equal-loss balancing is chosen as the target of reweighting.
>
> Please let us know if you have any further questions!

---

> > ### Author Rebuttal · Reviewer_jcbC · 2026-04-01
> >
> > Thank you for your detailed rebuttal and additional experiments. My concerns are partially resolved. I have a follow-up question: For Figure 2, have you compared your method with approaches motivated by NC, for example INC [1], to evaluate which methods produce a better NC evolution? To my understanding, the INC method regularizes more strongly toward a balanced ETF structur. However, it achieves worse accuracy than the proposed method. What is the author's perspective on this?
> >
> > [1] Liu, X., Zhang, J., Hu, T., Cao, H., Yao, Y., and Pan, L. Inducing neural collapse in deep long-tailed learning. AISTATS 2023
> >
> > **Update after receiving the authors’ “Reply Rebuttal Comment”:** Thank you for the further explanations and experiments. In my opinion, this discussion and the experiments with methods such as INC should be included in the paper. The accuracy drop after integrating INC with CE in your new experiment is interesting, and it raises the question of whether this method enforces over-regularization that is actually harmful to performance, and whether simply using the proposed loss balancing (which is less regularized) is preferable. This may require a closer look.
> >
> > Overall, my previous concerns are mostly resolved, and I will increase my score.

---

> > > ### Author Response · Authors · 2026-04-01
> > >
> > > We thank the reviewer for this important follow-up question.
> > >
> > > **Additional Experiments for INC.** We would like to first clarify that our paper already compared against NC-motivated approaches in the main experiments. In particular, we reported accuracy comparisons with INC-DRW, and we also used ETF-DR, another NC-motivated method, as a baseline to test whether our reweighting can further improve such approaches. However, we agree that the comparison for NC evolution was missing, so we conducted additional experiments.
> > >
> > > The additional result about NC metrics shows that our method achieves lower NC1, NC2 and NC3 than INC (see nc1\_inc\_ours.png, nc2\_inc\_ours.png, nc3\_inc\_ours.png). Moreover, we observe that both INC and our method perform well on NC1 metric, which matches the idea in [1] that NC1 is usually the easiest NC property to emerge during training. However, the result on the NC2 metric on INC may be a little different from our expectation. This may stem from a mismatch between the object explicitly regularized by INC and the object measured by our NC2 metric. In INC, the explicit NC2 regularization is imposed on the angular structure of the centered class means, whereas the NC2 metric we report follows [2] based on the classifier-weight geometry. Therefore, these quantities are related but not identical, and directly regularizing class-mean angles does not necessarily guarantee the best value under the NC2 metric we report.
> > >
> > > Furthermore, we conduct experiments on the accuracy side. As the result shown in the table (see acc\_inc\_ours.png), our method consistently outperforms CE+INC and CE+INC+Ours further improves over CE+INC, showing that our method is not only stronger on its own but also acts as a general plug-in reweighting module that can be combined with other methods, including INC-style regularization.
> > >
> > > **Perspective on INC versus our inverse reweighting.** Our perspective is that the difference between INC and our method is not simply about which one is more NC-like but about which one intervenes on the more fundamental control variable in long-tailed learning. INC mainly operates at the feature level, which explicitly regularizes the representation geometry so that the learned features and class means become more consistent with NC. In contrast, our method treats the simplex ETF geometry as an end-state reference rather than a direct optimization target and takes class-wise loss balancing as the ideal target that reweighting should explicitly optimize in long-tailed training.
> > >
> > > As discussed in our paper, an NC-consistent ETF solution implies equal per-class average loss, while persistent loss imbalance prevents convergence to such an ETF-consistent regime in Theorem 3. Therefore, in our view, NC should not be treated as a direct optimization target but rather as an end-state reference that reveals what reweighting should pursue. This is why we choose to directly optimize equal per-class average loss and formulate reweighting as an inverse problem with a closed-form per-class solution in Theorem 4, instead of heuristically imposing geometric constraints in feature space. Differently, INC intervenes on geometry, whereas our method intervenes on the imbalance mechanism that obstructs NC in the first place. From this perspective, the more critical issue in long-tailed training is not to directly prescribe what the feature geometry should look like, but to first correct the class-wise optimization imbalance that drives the training process away from a balanced solution. Then the geometric improvement is then a consequence of this correction rather than the sole objective. Results are shown in https://anonymous.4open.science/r/anonymous-repo-A24D
> > >
> > > Hope our reply can solve your problem.
> > >
> > > [1] Dang, H., Tran, T., Nguyen, T., and Ho, N. Neural collapse for cross-entropy class-imbalanced learning with unconstrained relu feature model. ICML 2024
> > >
> > > [2] Zhou, J., You, C., Li, X., Liu, K., Liu, S., Qu, Q., and Zhu, Z. Are all losses created equal: A neural collapse perspective. NeurIPS 2022
> > >
> > > **Update Comment**
> > >
> > > Thanks for the thoughtful feedback and updated score.
> > >
> > > Regarding the final question, we agree that over-regularization may plausibly explain the behavior of INC. NC-inspired methods such as INC and ETF-DR impose stronger geometric constraints by directly pulling the features or classifier toward a simplex-ETF structure, whereas our method does not explicitly enforce the geometry. As discussed in our paper, persistent loss imbalance under long-tailed training prevents convergence to such an ETF-consistent regime, so our method instead operates at the mechanism level by correcting this upstream imbalance, which may be a milder and more task-adaptive intervention and thus more favorable to performance.
> > >
> > > We sincerely appreciate this insightful discussion. If accepted, we will incorporate the feedback from the rebuttal into the revised manuscript and further improve its clarity and presentation.

---

### Official Review · Reviewer_pXv5 · 2026-03-10

**Soundness:** 2
**Presentation:** 3
**Significance:** 2
**Originality:** 2
**Overall Recommendation:** 4
**Confidence:** 4

**Summary:**

This paper proposes an inverse-view reweighting strategy for imbalanced learning, which helps learned embeddings exhibit the neural collapse phenomenon, known as an ideal representation. The authors show that the proposed method consistently outperforms other baseline methods in imbalanced learning.

**Compliance With Llm Reviewing Policy:**

Affirmed.

**Final Justification:**

The authors rebuttal clearly addressed my main concerns and improved my understanding of the work, leading me to increase my score.

**Key Questions For Authors:**

1. Could you provide more details about the Tikhonov regularizer? Why is this term necessary? What happens if $\alpha = 0$?
2. Why is it desirable for classifiers to exhibit the neural collapse phenomenon in long-tailed learning? One possible motivation is that classifiers trained on long-tailed datasets typically show inferior performance compared to those trained on balanced datasets, so it may be desirable to make long-tailed learning behave more similarly to balanced learning. However, it is not clear whether the representation obtained from balanced training is necessarily optimal in terms of performance. For example, suppose that NC1, NC3, and NC4 hold (in particular, the within-class variance is zero for all classes), but NC2 does not hold, while the class means remain sufficiently separable. In such a case, perfect classification may still be possible. For instance, if the datasets are arranged as illustrated in Figure 1, both the baseline and the proposed method could achieve perfect classification.

**Limitations:**

Although the authors do not explicitly discuss the limitations or potential negative societal impacts of their work, such a discussion does not seem necessary for this paper.

**Strengths And Weaknesses:**

Strengths:
1. The paper is well written and very easy to follow.
2. The authors provide proper theory that support their method.
3. They present formal equations for key concepts (such as simplex ETF and neural collapse), which make them easy for readers to understand.

Weaknesses:
1. Although authors argue that 'reweighting is often designed as a heuristic' in line 42, I do not fully agree with this statement. The authors argue that the training method should 'recover the ideal ETF geometry under a long-tailed scenario.' However, as Zhu et al. [a] show, modifying a constant used for averaging (which can be viewed as a form of reweighting) can construct the simplex ETF geometry. Moreover, since image data can be augmented using mixup, one can make the effective sample size equal across classes, which may allow construction of the simplex ETF geometry (e.g., SMOTE [b]). If the goal is to construct the simplex ETF, I believe the two methods mentioned above may be easier to apply than the proposed inverse-view reweighting strategy. Since this is my main concern, I would be willing to increase my score if this concern arises from a misunderstanding on my part.
2. Although authors argue that their method 'consistently improves performance across diverse baselines' in line 303, the results in Table 2 and Table 3 do not seem to fully support this claim. For example, 'ETF-DR+DisA' shows better performance than the proposed method on CIFAR-10-LT.


[a] Zhu, Jianggang, et al. Balanced contrastive learning for long-tailed visual recognition.CVPR. 2022.
[b] Chawla, Nitesh V., et al. SMOTE: synthetic minority over-sampling technique. Journal of artificial intelligence research 16 (2002): 321-357.

---

> ### Author Rebuttal · Authors · 2026-03-31
>
> We thank the reviewer for comments.
>
> **W1: The concern on ETF construction and alternative routes.**
>
> We believe there may be a misunderstanding of our intended claim. Our goal is not to directly construct simplex ETF geometry, but to study the loss reweighting problem itself and the ideal target for reweighting. Our main contribution is to make this target explicit, with NC serving only as the theoretical motivation and end-state reference. Therefore, our contribution is not ETF construction, but rethinking loss reweighting through an explicit target. From this perspective, the reviewer’s examples are valid but operate at different design levels: BCL promotes regular-simplex formation through contrastive learning, while SMOTE changes the effective sample distribution at the data level. In contrast, our work operates at the objective level and focuses on how class weights should be determined. Hence, these methods are not direct substitutes for the question studied in our paper.
>
> **W2: ETF-DR+DisA has better performance on CIFAR-10-LT.**
>
> We agree that "consistently improves" may be too strong, since there are exceptions such as ETF-DR+DisA on CIFAR-10-LT. Our point is the overall effectiveness across diverse baselines and settings, rather than superiority in every case. We also note that CIFAR-10-LT is a saturated benchmark, so more informative evidence comes from more challenging datasets such as ImageNet-LT. We will revise the wording accordingly.
>
> **KQ1: Details about the Tikhonov regularizer**
>
> The Tikhonov term is a tunable regularizer. It keeps the inferred class weights close to a prior weight $w_c^{(0)}$, making the inverse solution compatible with the baseline weighting scheme. When $\alpha=0$, the objective reduces to pure inverse loss equalization without any prior constraint. Thus, the term is not strictly necessary for the inverse formulation itself, but mainly serves as a practical regularizer for stability and compatibility. Our sensitivity study shows that $\alpha=0$ achieves the best accuracy on the CE baseline, while the best choice can vary across baselines. This suggests that the regularizer mainly provides a flexible way to incorporate prior weighting when desired, rather than being necessary for the method to work. In addition, the training loss and validation accuracy curves (see loss\_acc\_curve.png) show stable optimization and consistently better performance throughout training.
>
> Results are shown in https://anonymous.4open.science/r/anonymous-repo-A24D
>
> **KQ2: Why is Neural Collapse desirable in long-tailed learning?**
>
> Neural Collapse is desirable in long-tailed learning not because it simply resembles balanced training, but because it induces a more regular class geometry and more uniform inter-class separation, which improves robustness to small residual intra-class perturbations, especially for tail classes. We agree that NC2 is not necessary for perfect classification in the zero-noise limit. The key issue, however, is not noiseless separability, but optimal small-noise robustness. The following theorem formalizes this point.
>
> **Theorem (NC2 is unnecessary for noiseless separability but necessary for optimal small-noise robustness).**
> Consider the NC1 small-variance surrogate model
>
> $$
> h=\\mu_\\gamma+z,\\qquad z\\sim\\mathcal N(0,\\sigma^2 I),\\qquad \\gamma\\in\\{1,\\dots,C\\},
> $$
>
> with a self-dual classifier
>
> $$
> w_c=\\alpha\\mu_c,\\qquad b_c=-\\frac{\\alpha}{2}\\|\\mu_c\\|_2^2,\\qquad \\alpha>0.
> $$
>
> Let
>
> $$
> \\Delta(M):=\\min_{c\\neq c'}\\|\\mu_c-\\mu_{c'}\\|_2.
> $$
>
> Then
>
> $$
> \\beta(M):=\\lim_{\\sigma\\to 0}-\\sigma^2\\log\\Pr(\\hat\\gamma(h)\\neq\\gamma)
> =\\frac{1}{8}\\Delta(M)^2.
> $$
>
> Moreover, under $\\|\\mu_c\\|_2\\le 1$,
>
> $$
> \\beta(M)\\le \\beta^\\star:=\\frac{C}{4(C-1)},
> $$
> with equality if and only if the class means form a simplex ETF. **Therefore, NC2 is not necessary for perfect classification at $\\sigma=0$, but it is necessary for achieving the optimal error exponent under small noise.**
>
> **Proof sketch.**
> For class $c$, let
>
> $$
> s_c(h)=(w_c)^\\top h+b_c.
> $$
>
> For $h=\\mu_c+z$, the score difference is
>
> $$
> s_{c'}(h)-s_c(h)=\\alpha\\left((\\mu_{c'}-\\mu_c)^\\top z-\\frac{1}{2}\\|\\mu_c-\\mu_{c'}\\|_2^2\\right).
> $$
>
> Hence, misclassifying $c$ as $c'$ is equivalent to a one-dimensional Gaussian tail event, giving
>
> $$
> \\beta_{cc'}=\\frac{1}{8}\\|\\mu_c-\\mu_{c'}\\|_2^2.
> $$
>
> Therefore, the overall exponent is governed by the easiest confused pair:
> $$
> \\beta(M)=\\min_{c\\neq c'}\\beta_{cc'}=\\frac{1}{8}\\Delta(M)^2.
> $$
> By simplex ETF optimality under $\\|\\mu_c\\|_2\\le 1$, this exponent is maximized if and only if the class means form a simplex ETF.
>
> **Corollary.** If $\\Delta(M)>0$, NC2 is unnecessary for perfect classification at $\\sigma=0$; in the equal-norm case, NC2 failure implies $\\beta(M)<\\beta^\\star$.
>
> **Therefore, NC2 remains uniquely optimal in the large-deviation error exponent sense, which is why Neural Collapse is desirable in long-tailed learning.**

---

> > ### Author Rebuttal · Reviewer_pXv5 · 2026-04-01
> >
> > Thank you for your detailed response and for the clear explanation. My questions have been resolved, and I have accordingly increased my score. I hope that the discussed points will be well reflected in the revised version.

---

> > > ### Author Response · Authors · 2026-04-04
> > >
> > > Thank you for your thoughtful feedback and constructive suggestions.
> > >
> > > --------Overall response: Thanks to Reviewer pXv5 for the careful reading and constructive discussion. --------
> > >
> > > We sincerely thank Reviewer pXv5 for insightful comments. We are glad that our rebuttal helped clarify the reviewer's main concerns including Soundness, Significance, Originality and Presetation.
> > >
> > > For **Soundness**, (the justification for Neural Collapse), we provided theorem-based clarification showing that NC2, while unnecessary for zero-noise separability, remains optimal from the small-noise robustness and large-deviation error exponent perspective.
> > >
> > > For **Significance**, (the role of the Tikhonov regularizer), we clarified that it is not necessary for the method itself, but serves as a practical compatibility term, while $\alpha=0$ reduces to pure inverse loss equalization.
> > >
> > > For **Originality**, (the novelty beyond direct ETF construction), we clarified that our method is not designed to directly construct simplex ETF geometry, but to study loss reweighting itself and make its target explicit, with NC serving as an end-state reference rather than a direct optimization target.
> > >
> > > For **Presentation**, (the scope of the empirical claim), we acknowledged the point and will revise the wording in the final manuscript.
> > >
> > > We believe the rebuttal has addressed the reviewer's concerns and further strengthened the conceptual and theoretical aspects of the paper. While our method is motivated by the Neural Collapse view, we emphasize that our contribution lies in making the target of loss reweighting explicit and reformulating reweighting as an inverse problem, rather than direct ETF construction. We believe our work provides an alternative perspective on long-tailed learning.
> > >
> > > Overall, we thank the reviewer pXv5 for the positive recognition and valuable suggestions, and **we sincerely wish you success with your submissions and future work.**

---

### Official Review · Reviewer_YRWX · 2026-03-11

**Soundness:** 3
**Presentation:** 3
**Significance:** 3
**Originality:** 3
**Overall Recommendation:** 4
**Confidence:** 3

**Summary:**

This paper studies loss reweighting strategies for long-tailed classification, where class imbalance causes models to favor head classes over tail classes. The authors revisit reweighting from the perspective of **Neural Collapse (NC)**, which characterizes the geometric structure of features and classifiers at the terminal phase of training. Motivated by the observation that the ideal NC geometry implies **equal average loss across classes**, the paper formulates loss reweighting as an **inverse problem** whose goal is to infer class weights that achieve this equal-loss objective. Based on this view, the authors propose a dynamic reweighting method that estimates class weights during training to reduce loss imbalance and encourage alignment with the NC structure. The method is evaluated on several long-tailed classification benchmarks and compared with existing reweighting and imbalance learning approaches. Experimental results show improvements over strong baselines and demonstrate better alignment with Neural Collapse metrics.

**Compliance With Llm Reviewing Policy:**

Affirmed.

**Key Questions For Authors:**

1. **Justification of the equal-loss objective.**
   The method is motivated by the observation that the Neural Collapse regime suggests balanced class-wise losses. Could the authors further clarify the theoretical or empirical justification for enforcing equal average loss across classes during training? In particular, under what conditions is this objective expected to improve generalization in long-tailed settings?

2. **Relation to existing adaptive reweighting methods.**
   Several prior works propose dynamic or loss-based weighting strategies for imbalanced learning. Could the authors clarify more explicitly how the proposed inverse formulation differs from these approaches in practice? For example, are there specific properties or behaviors that arise uniquely from the proposed formulation?

3. **Stability and sensitivity of the weighting mechanism.**
   Since the proposed method dynamically estimates class weights during training, how sensitive is the approach to hyperparameters or noise in early training stages? An ablation study analyzing the stability of the estimated weights over training would help better understand the robustness of the method.

4. **Generality across architectures and imbalance regimes.**
   The experiments are conducted on standard long-tailed benchmarks. Could the authors comment on how well the method generalizes across different architectures, imbalance ratios, or datasets beyond those considered in the paper?

5. **Neural Collapse analysis.**
   The paper uses NC-related metrics to support the motivation of the method. Could the authors further analyze whether enforcing balanced class losses consistently leads to stronger alignment with NC properties across different datasets and training regimes?

**Limitations:**

Yes. The authors discuss the main limitations of their approach, including the scope of the experimental evaluation and the assumptions underlying the proposed formulation. They also acknowledge that the method is primarily evaluated in standard long-tailed classification settings and that further validation across broader datasets and scenarios would be useful. No direct negative societal impacts are apparent beyond those generally associated with machine learning systems trained on biased or imbalanced datasets.

**Strengths And Weaknesses:**

Strengths

Soundness.
The paper addresses the problem of long-tailed classification through the lens of Neural Collapse (NC), providing a principled motivation for revisiting loss reweighting methods. The authors argue that the NC regime suggests that the average loss across classes should be balanced, and they formulate loss reweighting as an inverse problem that aims to recover class weights enforcing this property. The proposed method dynamically adjusts class weights during training to reduce class-wise loss imbalance. The empirical evaluation is conducted on standard long-tailed classification benchmarks and includes comparisons with established baselines in the imbalance learning literature. The authors also analyze NC-related metrics to support the geometric motivation behind the method.

Presentation.
The paper follows a logical structure, starting with a discussion of class imbalance and existing reweighting approaches, then introducing the Neural Collapse perspective and the inverse formulation of the problem. The high-level motivation is intuitive and connects well to recent theoretical analyses of deep networks. The experimental section is clearly organized and presents results in a systematic way. However, some technical details—particularly the derivation of the inverse formulation and the implementation of the dynamic reweighting procedure—could be explained more clearly to improve readability and reproducibility.

Significance.
Class imbalance and long-tailed recognition are important problems in machine learning, particularly for real-world datasets where class distributions are naturally skewed. Reweighting methods remain widely used due to their simplicity and compatibility with standard training pipelines. By connecting reweighting strategies to Neural Collapse properties, the paper offers a new perspective that may help better understand the behavior of models trained on imbalanced data. This perspective could potentially inspire further research exploring geometric properties of deep representations for imbalance learning.

Originality.
The main novelty lies in interpreting loss reweighting from the perspective of Neural Collapse and formulating it as an inverse problem that aims to equalize class-wise losses. While both Neural Collapse analysis and loss reweighting methods have been studied extensively in prior work, the paper combines these ideas in a new way and derives a dynamic weighting scheme motivated by this perspective. The contribution is therefore primarily conceptual and methodological, providing an alternative interpretation of reweighting strategies.

Weaknesses

Soundness limitations.
Although the Neural Collapse perspective provides an interesting motivation, the theoretical justification connecting equal class-wise loss to improved long-tailed generalization is not fully established. Neural Collapse characterizes asymptotic geometric properties of trained networks, but it is not entirely clear that enforcing balanced losses during training necessarily leads to better performance in imbalanced settings. Additional theoretical discussion or analysis could strengthen this connection.

Experimental limitations.
The experimental evaluation is limited to a set of standard long-tailed benchmarks. While the method shows improvements over several baselines, the gains appear relatively modest in some settings. Additional experiments on more datasets, architectures, or imbalance scenarios would help demonstrate the robustness and general applicability of the approach. Further ablation studies analyzing the behavior of the dynamic weighting mechanism would also strengthen the empirical evidence.

Originality limitations.
Although the inverse formulation is an interesting conceptual contribution, the resulting algorithm is still closely related to existing adaptive reweighting strategies used in long-tailed learning. As a result, the practical novelty compared to prior dynamic weighting methods may appear incremental. A clearer comparison with closely related reweighting techniques would help highlight the distinctive aspects of the proposed approach.

Presentation limitations.
Some parts of the technical development are relatively dense, and the intuition behind certain design choices could be explained more clearly. Providing additional explanations or implementation details would make the method easier to understand and reproduce.

---

> ### Author Rebuttal · Authors · 2026-03-31
>
> We thank the reviewer for these comments.
>
> **W1 and KQ1: Justification of the equal-loss objective.**
>
> As we clarified in Reviewer ohLs (W2), equal loss is a principled target for loss reweighting inspired by Neural Collapse (simplex ETF geometry). And we also respond that the simplex ETF is beneficial for long-tailed learning because it provides a more desirable class geometry and is optimal in the large-deviation error exponent sense in Reviewer pXv5 (KQ2). Moreover, in response to Reviewer jcbC (KQ1), we clarified that ETF serves as the geometric end-state reference, while equal per-class average loss is the corresponding loss-level property that reweighting can directly optimize. This objective is expected to improve long-tailed generalization precisely when persistent class-wise loss imbalance and cross-batch optimization skew are key obstacles to approaching a more NC-consistent geometry. In that regime, equal-loss balancing is not only a necessary loss-level property of the desired end state, but also a directly controllable objective for reweighting.
>
> **W2 and KQ4: Generality across architectures and imbalance regimes.**
>
> We would like to clarify that our empirical evaluation is not limited to a single benchmark setting. Our conducted experiments already include CIFAR-10-LT, CIFAR-100-LT, ImageNet-LT, and iNaturalist, together with multiple imbalance ratios on CIFAR-LT and comparisons across diverse baselines. In addition, we also provide mechanism analysis beyond final accuracy, including NC metrics, t-SNE visualization, and the ablation study in Section 5. Therefore, we believe the current results already provide reasonable evidence that the method generalizes across standard long-tailed benchmarks. And if needed, we are willing to add an additional experiment on another backbone in the revision.
>
> **W3 and KQ2: Relation to existing adaptive reweighting methods.**
>
> Existing adaptive reweighting methods are not merely that the weights are dynamic and target-agnostic. In contrast, our method is derived from an explicit target rather than directly prescribed. Specifically, inspired by Neural Collapse, we first define equal effective class-wise average loss as an ideal target for loss reweighting and then solve an inverse problem to derive the corresponding class weights. This formulation further yields a closed-form batch-wise solution, which makes the resulting weighting mechanism explicit and interpretable. As clarified in our response to Reviewer ohLs (W3), our reweighting method is not heuristic-designed but target-driven and lightweight.
>
> **W4: Presentation limitations.**
>
> The paper already provides the inverse formulation, its closed-form solution, and the complete training algorithm. We also include visualization and mechanism-analysis results, such as NC metrics and t-SNE plots, to support the method’s design intuition. We hope our responses to the key questions further clarify the intuition, formulation, and practical differences of the method.
>
> **KQ3: Stability and sensitivity of the weighting mechanism.**
>
> Regarding stability and sensitivity, our current manuscript already contains partial evidence through the sensitive analysis of the hyperparameter for the Tikhonov regularizer in the appendix. The results indicate that the method is not tied to a single brittle hyperparameter setting and remains reasonably stable under different choices of the regularization strength. Furthermore, we conduct additional experiments to directly analyze the behavior of the estimated weights over training. Specifically, we visualize the trajectories of representative head, medium, and tail class weights, showing that the estimated weights remain bounded throughout training while maintaining a clear hierarchy, with head classes consistently downweighted and tail classes consistently upweighted (see weight\_trajectories.png). The epoch-to-epoch variation further shows that these dynamic adjustments are well controlled and become progressively more stable over training (see weight\_delta.png).
>
> **KQ5: Neural Collapse analysis.**
>
> We further evaluated NC2 and NC3 metrics (see nc_metrics.png) on both CIFAR-10-LT and CIFAR-100-LT under multiple imbalance factors (20, 50, 100, and 200), comparing the standard CE baseline with CE + our method. The additional results show a clear overall trend that our method achieves lower NC2 and NC3 values than the CE baseline across these datasets and imbalance regimes, indicating stronger alignment with NC geometry. Due to time limitations, we have not extended this additional NC analysis to iNaturalist and ImageNet-LT. Nevertheless, this result already supports the conclusion that our method leads to stronger alignment with NC properties across different datasets and imbalance factors.
>
> Results are shown in https://anonymous.4open.science/r/anonymous-repo-A24D
>
> Please let us know if you have any further questions!

---

> > ### Author Rebuttal · Reviewer_YRWX · 2026-04-02
> >
> > Thank you for the detailed and constructive rebuttal. The additional clarifications are helpful and address several of my concerns.
> >
> > In particular, the explanation of the equal-loss objective as a loss-level counterpart to the Neural Collapse (ETF) geometry provides a clearer conceptual justification for the proposed formulation. The distinction between target-driven reweighting and heuristic adaptive weighting methods is also better articulated, especially with the inverse problem perspective and closed-form solution.
> >
> > I also appreciate the additional experimental details regarding stability and sensitivity, including the analysis of the Tikhonov regularizer and the visualization of weight trajectories. These results help clarify that the proposed method is reasonably stable during training. Similarly, the extended evaluation of NC metrics (NC2 and NC3) across multiple imbalance factors strengthens the empirical support for the claimed alignment with Neural Collapse properties.
> >
> > The clarification on the breadth of experimental evaluation (including CIFAR-LT, ImageNet-LT, and iNaturalist) is helpful, although additional results on more diverse architectures would further strengthen the generality claims.
> >
> > Overall, the rebuttal improves the clarity of the motivation and provides additional empirical evidence supporting the proposed approach. While some questions regarding the theoretical grounding and broader generalization remain, my confidence in the paper has increased.

---

> > > ### Author Response · Authors · 2026-04-04
> > >
> > > Thank you for your thoughtful feedback and constructive suggestions.
> > >
> > > --------Overall response: Thanks to Reviewer YRWX for the careful reading and constructive discussion. --------
> > >
> > > We sincerely thank Reviewer YRWX for the insightful feedback. In the rebuttal, we carefully addressed the main concerns on Soundness, Significance, Originality, and Presentation.
> > >
> > > For **Soundness**, (the justification of the equal-loss objective and the stability of the weighting mechanism), we clarified that equal per-class average loss is the directly optimizable loss-level counterpart of the Neural Collapse end-state reference, and further strengthened the empirical support with additional stability, sensitivity, and NC analyses.
> > >
> > > For **Significance**, (the generality across datasets and imbalance regimes), we clarified that our evaluation already covers multiple standard long-tailed benchmarks, multiple imbalance factors, and mechanism-level analyses beyond final accuracy.
> > >
> > > For **Originality**, (the distinction from existing adaptive reweighting methods), we clarified that our method is target-driven rather than target-agnostic, and derives class weights from an explicit inverse problem with a closed-form batch-wise solution.
> > >
> > > For **Presentation**, (the clarity of the intuition and formulation), we further clarified the relation between the Neural Collapse perspective and the equal-loss objective.
> > >
> > > We believe the rebuttal has strengthened the paper by clarifying both its conceptual motivation and its empirical support. In particular, we clarified the role of the equal-loss objective, the distinction between our target-driven inverse formulation and existing adaptive reweighting methods, and the additional evidence on stability, sensitivity, and NC-related behavior. We hope this helps highlight the paper's value as a more principled formulation of loss reweighting in long-tailed learning.
> > >
> > > Overall, we thank the reviewer YRWX for the positive recognition and valuable suggestions, and **we sincerely wish you success with your submissions and future work.**

---

### Official Review · Reviewer_ohLs · 2026-03-12

**Soundness:** 2
**Presentation:** 2
**Significance:** 2
**Originality:** 3
**Overall Recommendation:** 4
**Confidence:** 3

**Summary:**

## Summary:
This paper studies loss reweighting for long-tailed classification. It proposes class-wise equal average loss as the target condition, motivated by a Neural Collapse perspective under ideal simplex ETF geometry, and formulates reweighting as a regularized inverse problem with a closed-form dynamic class-weighting solution.

The practical method combines a batch-wise inverse weighting term with a macro-level frequency-aware factor to address both within-batch loss imbalance and across-batch class occurrence imbalance. Experiments are conducted on CIFAR-100-LT, CIFAR-LT, ImageNet-LT, and iNaturalist, with additional analyses including loss imbalance, Neural Collapse related metrics, and ablations.

**Compliance With Llm Reviewing Policy:**

Affirmed.

**Final Justification:**

After careful consideration, the overall response to the question is relatively comprehensive.

**Key Questions For Authors:**

1.This strategy seems potentially transferable to other tasks as well. What exactly makes the method specifically suited to long-tailed learning?
More concretely, if the proposed idea can also be applied to other imbalance-related settings, what is the key task-specific insight that makes it particularly relevant or necessary for long-tailed classification?

2.The main experimental evidence is largely based on attaching the proposed component to other methods and showing improved performance. I am curious about the parameter cost of doing so. How many additional parameters are introduced when the method is combined with these baselines? Since performance gains can also come from increased model capacity or additional components, it would be helpful for the authors to clarify how they isolate the effectiveness of the proposed method itself, and whether the improvements are achieved with minimal additional parameter overhead.

**Limitations:**

yes

**Strengths And Weaknesses:**

### Strengths:

1.The paper provides a Neural Collapse (NC) based perspective for understanding class-wise loss imbalance in long-tailed learning, together with a relatively complete theoretical derivation.
By connecting long-tailed loss imbalance to the deviation from the ideal NC geometry, the paper offers a more principled interpretation of why reweighting may be needed.

2.The proposed method is relatively simple, and the empirical evaluation is fairly extensive.
The method itself is lightweight and easy to integrate into existing pipelines, and the paper includes experiments across multiple settings along with auxiliary analyses.

### Weaknesses:

1.The problem awareness is not particularly new.
The idea that training signals across classes should be balanced under class imbalance has long been one of the central themes in long-tailed reweighting. In this sense, the paper appears to provide a new interpretation of an existing issue, rather than identifying a fundamentally new problem.

2.The theoretical part feels more supportive than decisive, and the macro-level component still appears heuristic.
While the theory helps motivate the method, it does not seem to constitute a decisive theoretical breakthrough. In particular, the macro-level design still has a noticeable heuristic flavor, which weakens the sense that the full method is derived in a fully principled manner.

3.The method itself does not appear to introduce a particularly large methodological innovation.
Although the formulation is clean and the presentation is well structured, the core technical novelty of the method seems somewhat limited.

---

> ### Author Rebuttal · Authors · 2026-03-31
>
> We thank the reviewer for these comments.
>
> **W1: The problem awareness is not particularly new.**
>
> We do not identify the idea that training signals across classes should be balanced under class imbalance as a new problem. Instead, we view balance as the desired target of reweighting, which is inspired by Neural Collapse. Our point is that many prior reweighting methods give class weights in a heuristic manner and are target-agnostic. In contrast, we formalize balance as an explicit and ideal loss target for reweighting. We formulate equal per-class average loss as the NC-desired end-state objective and treat reweighting in an inverse view that derives class weights from this target. Therefore, our contribution lies not in identifying a new problem, but in providing a principled target and a target-driven formulation for reweighting.
>
> **W2: The theory is supportive, while the macro-level part appears heuristic.**
>
> We respectfully believe the theoretical part plays a more decisive role in our method than merely providing post-hoc support. In our paper, the theory not only motivates the method but also identifies the explicit and ideal target for loss reweighting. Specifically, we show that class-wise average loss must equal and further prove that persistent loss imbalance prevents convergence to such an ideal solution in Theorem 3, which gives an explicit and concrete target for reweighting. Based on this, we treat reweighting as an inverse problem and derive a closed-form per-class solution in Theorem 4.1. Therefore, the theoretical part in our work is not merely supportive but decisive in specifying the target of reweighting and deriving the core batch-wise method.
>
> Regarding the macro-level component, it is introduced to address a long-tailed training dynamics issue explicitly described in Section 4.2-4.3, where head classes appear in far more mini-batches than tail classes, so batch-wise reweighting alone cannot fully remove the cumulative optimization imbalance across training. This is why the macro-level term is particularly relevant to long-tailed learning rather than being an arbitrary heuristic, and its effectiveness is further supported by the ablation in Section 5.4.
>
> **W3: The methodological novelty appears limited.**
>
> We would like to clarify that the methodological novelty of our work is not in introducing a more complex weighting formula or an additional large module. Instead, our method first specifies an explicit end-state equal class-wise average loss motivated by Neural Collapse, which is different from many existing reweighting methods that typically prescribe class weights directly. Moreover, we treat reweighting as an inverse problem to derive class weights from this target and lead to a closed-form per-class batch-wise solution. Therefore, the technical novelty of our method lies not in added architectural complexity, but in a new target-driven lightweight solution principle for long-tailed reweighting.
>
> **KQ1: What makes the method specifically suited to long-tailed learning?**
>
> We agree that the inverse-target perspective may also be useful in other imbalance-related settings, and we view this transferability as a strength rather than a weakness. Inspired by Neural Collapse, our method drives equal per-class loss toward balance and encourages the model to approach a more NC-consistent regime to alleviate persistent class-wise imbalance in long-tailed learning. Moreover, our full method combines an NC-inspired inverse reweighting objective to mitigate long-tailed imbalance with a macro-level compensation to address long-tail-specific training dynamics.
>
> Empirically, this design is particularly relevant to long-tailed learning because our method consistently improves performance on multiple standard long-tailed benchmarks. We also conduct an additional visualization experiment, and the resulting weight trajectories (see weight\_trajectories.png) exhibit a clear head-medium-tail hierarchy throughout training, which directly reflects the intended head-suppression and tail-compensation behavior in long-tailed learning. More details about the extra visualization experiment can be seen in the response to Reviewer YRWX (KQ3).
>
> Results are shown in https://anonymous.4open.science/r/anonymous-repo-A24D
>
> **KQ2: Parameter cost and resource overhead.**
>
> Our method introduces no additional learnable parameters when combined with the baselines. As discussed in Appendix H, it only computes dynamic class weights in closed form, so the improvement comes from the reweighting mechanism itself rather than increased model capacity. To further clarify the resource overhead, we have also added an additional timing experiment. Under the standard CE baseline, the average time for 5 epochs is 8.45 s, while plus our method takes 8.79 s, corresponding to only a little increase.
>
> Please let us know if you have any further questions!

---

> > ### Author Rebuttal · Reviewer_ohLs · 2026-04-02
> >
> > My problem has been resolved, and the score has been improved. I hope the author will complete the content in the follow-up.

---

> > > ### Author Response · Authors · 2026-04-04
> > >
> > > Thank you for your thoughtful feedback and constructive suggestions.
> > >
> > > --------Overall response: Thanks to Reviewer ohLs for the careful reading and constructive discussion. --------
> > >
> > > We sincerely thank Reviewer ohLs for the insightful feedback. In the rebuttal, we carefully addressed the main concerns on Soundness, Significance, Originality, and Presentation.
> > >
> > > For **Soundness**, (the role of the theory and the macro-level component), we clarified that the theory is not merely supportive, but decisive in specifying the equal-loss target and deriving the closed-form batch-wise inverse solution, while the macro-level term addresses long-tail-specific cross-batch optimization imbalance.
> > >
> > > For **Significance**, (the relevance of the method to long-tailed learning), we clarified that our method directly targets persistent class-wise imbalance and long-tail training dynamics, and further showed that it introduces no additional learnable parameters and only minimal computational overhead.
> > >
> > > For **Originality**, (the novelty beyond existing balancing ideas), we clarified that our contribution does not lie in identifying balance itself as a new problem, but in making equal loss an explicit NC-inspired target and reformulating reweighting as a lightweight target-driven inverse problem.
> > >
> > > For **Presentation**, (the scope and positioning of the contribution), we further clarified the intended role of the theory, the task-specific relevance of the method, and the practical overhead of the full design.
> > >
> > > We believe the rebuttal has addressed the reviewer's concerns and further clarified the contribution and scope of the paper. In particular, we clarified the explicit equal-loss target, the decisive role of the theory, the long-tail-specific role of the macro-level compensation, and the lightweight nature of the method. Together, these points better highlight the paper's contribution as a more principled formulation of loss reweighting in long-tailed learning.
> > >
> > > Overall, we thank the reviewer ohLs for the positive recognition and valuable suggestions, and **we sincerely wish you success with your submissions and future work.**

---

### Decision · Program_Chairs · 2026-04-30

**Decision:**

Accept (regular)

**Comment:**

This paper studies long-tailed reweighting from a Neural Collapse perspective. It treats reweighting as an inverse problem and uses equal per-class average loss as the target. The method is simple, easy to plug into existing training pipelines, and is tested on several standard long-tailed benchmarks with additional analysis of NC-related behavior.

The post-rebuttal discussion is clearly positive. Three reviewers said their concerns were fully resolved after the rebuttal. They found the motivation clearer, especially the explanation of equal-loss as a loss-level counterpart to the ideal NC geometry, and they also found the added experiments helpful for understanding stability and behavior during training. The last reviewer initially asked for a more direct comparison with other NC-motivated methods, such as INC, but after the follow-up discussion and extra experiments, they also said that most of their concerns were resolved and that they would increase their score.

At this point, the remaining issues are small. The main suggestion from the discussion is that the paper would be even stronger if it included more of the new comparison and explanation around INC and related NC-based methods. I agree with that suggestion, but I do not see it as a reason to reject the paper. The current record supports that the method is sound, practically useful, and well supported by the experiments.

I therefore recommend acceptance. For the final version, I encourage the authors to include the added comparison with NC-motivated alternatives and the related discussion, since this will make the paper easier to understand and better positioned for readers interested in Neural Collapse.